# Evaluating multiple models using labeled and unlabeled data

## Abstract

It remains difficult to evaluate machine learning classifiers in the absence of a large, labeled dataset. While labeled data can be prohibitively expensive or impossible to obtain, *unlabeled* data is plentiful. Here, we introduce Semi-Supervised Model Evaluation (SSME), a method that uses both labeled *and* unlabeled data to evaluate machine learning classifiers. SSME is the first evaluation method to take advantage of the fact that: (i) there are frequently multiple classifiers for the same task, (ii) continuous classifier scores are often available for all classes, and (iii) unlabeled data is often far more plentiful than labeled data. The key idea is to use a semi-supervised mixture model to estimate the joint distribution of ground truth labels and classifier predictions. We can then use this model to estimate any metric that is a function of classifier scores and ground truth labels (e.g., accuracy or expected calibration error). We present experiments in four domains where obtaining large labeled datasets is often impractical: (1) healthcare, (2) content moderation, (3) molecular property prediction, and (4) image annotation. Our results demonstrate that SSME estimates performance more accurately than do competing methods, reducing error by $5.1\times$ relative to using labeled data alone and $2.4\times$ relative to the next best competing method. SSME also improves accuracy when evaluating performance across subsets of the test distribution (e.g., specific demographic subgroups) and when evaluating the performance of large language models.

## 1 Introduction

Rigorous evaluation is essential to the safe deployment of machine learning classifiers. The standard approach is to measure classifier performance using a large labeled dataset. In practice, however, labeled data is often scarce (Culotta & McCallum, 2005; Dutta & Das, 2023). Exacerbating the challenge of evaluation, the number of off-the-shelf classifiers has increased dramatically through the widespread usage of model hubs. The modern machine learning practitioner thus has a myriad of trained models, but little labeled data with which to evaluate them.

In many domains, *unlabeled data* is much more abundant than labeled data (Bepler et al., 2019; Sagawa et al., 2022; Movva et al., 2024). To take advantage of this, we introduce Semi-Supervised Model Evaluation (SSME), a method that can be used to evaluate multiple classifiers using both labeled *and* unlabeled data. Our key idea is to estimate the joint distribution of ground truth classes $y$ and continuous classifier scores $\mathbf{s}$ using a mixture model, where different components of the mixture model correspond to different classes. The joint distribution allows us to evaluate performance on examples where we have access *only* to each classifier's scores, ~~i.e., no labeled data.~~ i.e. unlabeled examples. SSME can estimate any metric which is a function of class labels and probabilistic predictions, which includes widely-used metrics like accuracy, expected calibration error, AUC, and AUPRC.

SSME is the first evaluation method to learn from three key facets of modern machine learning settings: (i) multiple machine learning classifiers, (ii) probabilistic predictions over all classes, and (iii) unlabeled data. Simultaneously using all three is difficult because it requires accurately estimating the (potentially high-dimensional) joint distribution $P(y, \mathbf{s})$ with primarily unlabeled data. While prior work captures subsets of these properties (Welinder et al., 2013; Platanios et al., 2017; Ji et al.,

2020; Chouldechova et al., 2022; Boyeau et al., 2024) — for example, augmenting labeled data with unlabeled data to evaluate a *single* classifier — no existing approach accommodates all three.

We show that using all available data — multiple classifiers, continuous scores over all classes, and unlabeled data — enables SSME to produce more accurate performance estimates compared to prior work. We test SSME's ability to estimate the absolute performance of each classifier across eight tasks, four metrics, and dozens of classifiers, where SSME accepts a set of classifiers, little labeled data (i.e. between 20 and 100 labeled examples) and more abundant unlabeled data (i.e. 1000 unlabeled examples). Concretely, we make four contributions:

1. We propose SSME, a method to evaluate multiple classifiers using labeled and unlabeled data. SSME extends readily to any number of classifiers and classes, and is able to estimate any metric that compares predicted probabilities to ground truth labels.
2. We conduct semi-synthetic experiments to characterize factors affecting SSME's performance: the accuracy, calibration, and cardinality of the classifier set being evaluated.
3. We show, in experiments spanning multiple modalities, domains, and classifier architectures, that SSME achieves the lowest metric estimation error compared to using labeled data alone and compared to prior work, across all considered metrics.
4. We demonstrate two broadly useful applications of SSME: evaluating subgroup-specific performance, a critical step in assessing algorithmic fairness, and evaluating fine-tuned large language models.

## 2 RELATED WORK

Our work builds on two areas of literature: methods that use a combination of labeled and unlabeled data to 1) evaluate a single classifier, or 2) evaluate the accuracy of multiple discrete annotations. For a discussion of connections to prediction-powered inference and unsupervised classifier evaluation, see Appendix A.

**Semi-supervised evaluation of single classifiers** involves the evaluation of a single classifier using both labeled and unlabeled data. There are two types of assumptions common in this literature. The first places parametric constraints on the distribution of classifier scores. Several works attempt to fit a mixture model to the distribution of classifier scores (Welinder et al., 2013; Chouldechova et al., 2022; Miller et al., 2018), as we do, while others apply techniques from Bayesian calibration (Ji et al., 2020; 2021). Our work differs in that the proposed framework naturally capitalizes on multiple classifiers, and as our results show, doing so results in improved estimates of performance. The second type of assumption relates to the structure of the shift between the labeled and unlabeled data; as Garg et al. (2022) establish, estimating accuracy on the unlabeled data is impossible absent assumptions about the nature of the distribution shift. Examples of these assumptions include covariate shift (Chen et al., 2021b; 2022; Lu et al., 2023), conditional independence of features (Steinhardt & Liang, 2016), and calibration on the unlabeled data (Guillory et al., 2021; Jiang et al., 2022). Here too, all work focuses on evaluating individual classifiers and often relies on larger amounts of labeled data than we assume (on the order of hundreds of labeled examples). In contrast, our focus is on the evaluation of *multiple* classifiers, when the amount of labeled data is too small to reliably learn any model of distribution shift between the labeled and unlabeled data.

**Semi-supervised evaluation of discrete annotators** was first introduced by Dawid & Skene (1979), who proposed a method to estimate ground truth in the presence of multiple potentially noisy discrete annotations. Many follow-on works inherit Dawid-Skene's strong assumption of class-conditional independence of annotator errors (Parisi et al., 2014; Platanios et al., 2017), including popular approaches in weak supervision (Ratner et al., 2017; Bach et al., 2017; Fu et al., 2020), where annotators are instead user-provided labeling functions. Such an assumption is plausible in certain contexts, but does not naturally translate to sets of candidate classifiers, whose predictions are likely to be correlated. Subsequent work has made an effort to relax the assumption of class-conditional independence, replacing it with independence conditional on a latent notion of example difficulty (Paun et al., 2018) or adjusting for dependencies between annotators (Ratner et al., 2017; Bach et al., 2017; Fu et al., 2020). However, these methods are designed to estimate the accuracy of *binary* annotations; they do not exploit the continuous probabilities available in multi-classifier evaluation. While some work has made progress towards accommodating

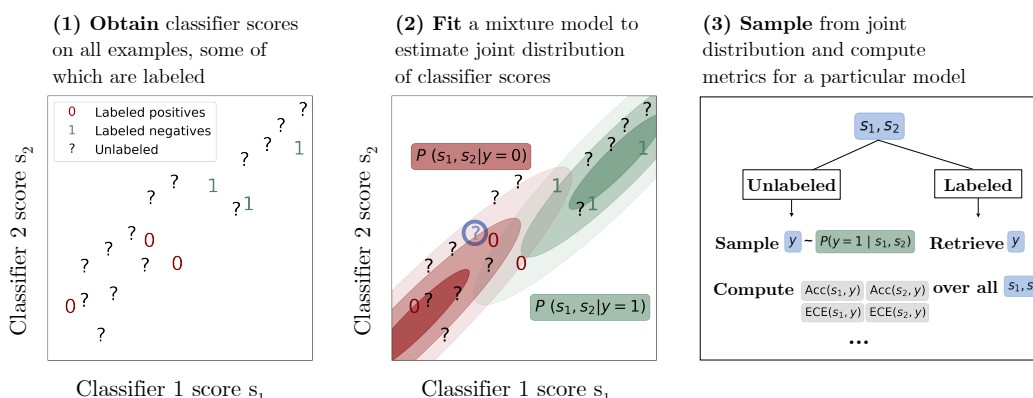

Figure 1: **Using SSME with two binary classifiers.** (1) Retrieve classifier scores on all examples, a small subset of which are labeled. (2) Fit a mixture model to estimate the joint density of scores and labels (where $s_1$ corresponds to the score assigned by the first classifier, and $s_2$ the second). (3) Use the resulting density to estimate metrics such as accuracy or expected calibration error. For instance, for the unlabeled blue point in panel 2, we sample a label $y$ several times (left); if the example were labeled (right), we would use the true $y$. SSME extends readily to any number of classifiers and classes $K$, and supports any metric that compares classifier scores to ground truth labels.

continuous predicted probabilities (Nazabal et al., 2016; Pirš & Štrumbelj, 2019), their focus is optimal aggregation, in contrast to our own, which is evaluation. Recent work uses a continuous notion of classifier confidence in conjunction with discrete annotations from each annotator (Goh et al., 2022; Boyeau et al., 2024), but do not use the distribution of classifier scores over *all* classes.

## 3 Problem Setting

We consider a setting in which a practitioner wishes to evaluate several classifiers. Formally, there are $M$ classifiers $[f_1, f_2, \ldots, f_M]$ designed for the same task. These classifiers may differ in their training data, function class, or training hyperparameters, among other possibilities.

Each classifier in the set maps from the same input domain $\mathcal{X}$ to a probability distribution over $K$ classes, i.e. $f_j : \mathcal{X} \to \Delta^{K-1}$. Let $\mathbf{s}^{(i)} = [f_1(x^{(i)}), f_2(x^{(i)}), \ldots, f_M(x^{(i)})]$ denote the concatenated set of classifier scores on a particular instance $x^{(i)}$.

During evaluation, we have access to the set of classifiers and two datasets: (1) a small labeled dataset, $\mathcal{D}_L = \{(x^{(i)}, y^{(i)})\}_{i=1}^{n_\ell}$ and (2) a larger unlabeled dataset $\mathcal{D}_U = \{(x^{(i)})\}_{i=1}^{n_u}$. The goal is to estimate classifier performance using metrics such as expected calibration error (ECE) or accuracy. If one knew the true label $y^{(i)}$ for each point $x^{(i)}$, it would be straightforward to evaluate the performance of each pre-trained classifier. However, in practice, the true label is not available for unlabeled examples, so we aim to infer (a distribution over) these labels. We assume in our setting that unlabeled data is far more available than labeled data, i.e. $n_u >> n_\ell$.

Such settings are common in applications of machine learning. In many domains, we have far more unlabeled data than labeled data: genomic variants outnumber our resources to experimentally reveal associations with disease (Sherman et al., 2022), and the amount of healthcare data exceeds our capacity to provide expert-adjudicated diagnoses (Movva et al., 2024).

## 4 Method

Our aim is to develop an approach that captures three common properties of modern classification settings: (i) multiple available trained models, (ii) an abundance of unlabeled data, relative to labeled data, and (iii) access to each classifier's predicted class probabilities on every input.

The core idea underlying SSME is that if we can estimate the joint distribution of ground truth labels $y$ and classifier scores $\mathbf{s}$ — i.e., $P(y, \mathbf{s})$ — then we can estimate any metric that is a function of classifier scores and ground truth labels. Notably, we use the classifier scores $\mathbf{s}$ (i.e. the probability outputs over all classes) and *not* the raw inputs $\mathbf{x}$ to estimate this joint density, consistent with prior semi-supervised evaluation work (Ji et al., 2020; Boyeau et al., 2024). There are two reasons for this: (1) the output distribution over classes $\mathbf{s}$ and the corresponding labels $y$ are sufficient to characterize most standard metrics (calibration, accuracy, etc.), and (2) modeling $s$ directly is a standard technique to avoid estimating densities over $x$, which can be difficult due to the frequently high-dimensional nature of the input distribution.

SSME makes two additional assumptions on the (true) mixture density $p(y, \mathbf{s})$. First, we assume that the unlabeled samples $\mathbf{s}^{(i)}$ are drawn from the same distribution as the labeled samples $(y^{(i)}, \mathbf{s}^{(i)})$. Second, given the challenges of density estimation in high-dimensional settings (Wang & Scott, 2019; Rippel & Adams, 2013), we primarily focus on settings with a limited ($\leq 50$) number of classes. Our latent variable model can support higher-dimensional evaluation problems but would require robust semi-supervised density estimation procedures.

To estimate the accuracy of the first model, for instance, one could (repeatedly) draw labels $\hat{y}$ from $P(y|\mathbf{s})$ for each datapoint $\mathbf{s}$ and compute agreement between the drawn label and the first model's predicted label. By modeling $P(y, \mathbf{s})$ using a mixture model, we can capture class-specific variation in model scores — for instance, classifiers in our set may agree in their predictions on $y = 0$ but disagree in their predictions on $y = 1$.

To estimate the joint distribution $P(y, \mathbf{s})$, we maximize the log likelihood over both the labeled and unlabeled datasets. When $y$ is unobserved, we treat it as a latent variable and marginalize it out. Overall, we maximize the following expression:

$$\max_{\theta} \log P(\mathbf{S}, Y, \mathbf{S}'; \theta) = \max_{\theta} \log \left[ \prod_{i=1}^{n_\ell} P(\mathbf{s}_i, y_i; \theta) \prod_{j=1}^{n_u} P(s_j; \theta)^{\lambda} \right]$$

$$= \max_{\theta} \sum_{i=1}^{n_\ell} \log \left[ P_\theta(\mathbf{s}_i|y_i) P_\theta(y_i) \right] + \lambda \sum_{j=1}^{n_u} \log \sum_{k=1}^{K} \left[ P_\theta(\mathbf{s}_j|y_j = k) P_\theta(y_j = k) \right]$$

where $\lambda_L$ modulates the relative weight of the labeled data in the likelihood. We fix $\lambda_L = 1$ in our main experiments; alternative weights are possible but require careful hyperparameter tuning or domain knowledge.

**Parametrization** Our approach can accommodate multiple parameterizations of the class-conditional distribution of scores, as long as they can be learned in the semi-supervised setting described above. We denote the parameterized distribution as $P_\theta(\mathbf{s}|y)$. Here, we use a kernel density estimator (KDE) to parameterize $P_\theta(\mathbf{s}|y)$. The KDE for the $k$th class can be written as:

$$P_\theta(\mathbf{s}|y = k) = \frac{1}{\sum_i P(y^{(i)} = k|s^{(i)})} \sum_i \mathcal{K}_h(\mathbf{s} - s^{(i)}) \cdot P(y^{(i)} = k|s^{(i)})$$

where the learnable parameters are the bandwidth $h$ and the kernel type $\mathcal{K}$. Note that unlike a traditional KDE, we weight each point by the probability it's in the cluster (i.e., a soft label). When $\mathbf{s}^{(i)}$ is labeled, $P(y^{(i)} = k) = 1$ for the true label $k$, and when $\mathbf{s}^{(i)}$ is unlabeled, we assign soft labels to each point.

Kernel density estimators are well-suited for the task, as they do not make parametric assumptions on the distributional form of each component; this is useful for modeling distributions of predictions, which can vary widely across outcomes and models.

Fitting densities on simplices (or more broadly bounded domains) results in biased estimates near the boundaries, a problem known as the boundary bias problem (Jones, 1993). To overcome this challenge, we utilize invertible *compositional data transforms*, the preferred method for analyzing data

with unit-sum constraints (Aitchison, 1982; Pawlowsky-Glahn & Buccianti, 2011). To implement this, we transform probabilistic predictions over $K$ classes to scores in $\mathbb{R}^{K-1}$ using the additive log ratio transform, which produces a one-to-one mapping between the two spaces. Additive log-ratio transforms are just a *reparameterization trick*; they simply transform the classifier scores from a hard-to-model space — the probability simplex — to an easier-to-model space — unbounded reals. For details on the additive log-ratio transform, see Appendix E.1. Our approach helps us overcome well-documented issues with density estimation over bounded spaces.

**Model estimation**    For all experiments, we fit each kernel density estimator with a Gaussian kernel. The kernel bandwidth is estimated using the improved Sheather-Jones algorithm (Botev et al., 2010).

In accordance with previous semi-supervised mixture models, we train our mixture model with expectation-maximization (EM) (Dempster et al., 1977; Zhao et al., 2023), which alternates between (1) the E-step, which estimates which mixture component each datapoint belongs to, and (2) the M-step, which estimates the parameters of each component based on the soft component assignments, including the overall class prior $P(y)$. We take the same approach, where we alternate between estimating the true label for a given example, and estimating the class-conditional distribution of model predictions based on these estimated labels. We optimize the parameters using EM over 50 epochs. We initialize component assignments by drawing a label for a given example according to the mean classifier score across the set of classifiers.

**Evaluation**

~~After fitting the mixture model, we sample the true label for each unlabeled point using our fitted posterior distribution for $P(y^{(i)}|\mathbf{s}^{(i)})$ multiple times for each unlabeled example $\mathbf{s}^{(i)}$ in our dataset. Alternatively, one could compute an expectation over the the label of each individual $\mathbf{s}^{(i)}$ and sum over the entire dataset; we discuss this approach in Appendix E.2. For labeled examples, we use the true $y^{(i)}$. The estimated metrics are computed by averaging across all realizations (sampled for unlabeled, fixed for labeled) of $y^{(i)}$.~~ Once we have fit the parameters $\theta$ for our mixture model, we can use our fitted density $P_\theta(\mathbf{s}, y)$ to estimate metrics of interest, using the procedure described in panel 3 of Figure 1. In particular, given an estimate of the full joint density $P_\theta(\mathbf{s}, y)$, we can infer a distribution over the label $y^{(i)}$ for every unlabeled example $\mathbf{s}^{(i)}$ and repeatedly draw ground truth labels from the distribution. To estimate the accuracy of classifier $j$, we would then compute, in expectation, how $\mathbf{s}_j^{(i)}$ agrees with $y^{(i)}$, and average this over all examples $i$. For unlabeled examples, we use the inferred distribution over $y^{(i)}$, and for labeled examples we use the true $y^{(i)}$. Details on our metric evaluation procedure are available in Appendix E.2. Finally, it may be the case that not all classifiers a practitioner wishes to evaluate are simultaneously available (for example, during active model development). In this case, one can refit SSME to the expanded classifier set.

**Alternative parameterizations**    SSME can also accommodate alternate parameterizations of $P(\mathbf{s}^{(i)}|y^{(i)})$ provided they can (1) accommodate both labeled and unlabeled data and (2) be fit using the mixture model framework described above. One alternative that we explore is using a normalizing flow to model each mixture component; we detail our approach and investigate when this improves over the KDE parameterization in Appendix C. While normalizing flows can also be learned in a semi-supervised setting, they often struggle to model multimodal distributions (Stimper et al., 2022; Cornish et al., 2020). We use the KDE to generate our results in binary datasets. We additionally test the normalizing flow in multiclass settings and show it performs competitively.

## 5 EXPERIMENTS

### 5.1 DATASETS AND CLASSIFIER SETS

We select datasets and classifier sets to be realistic and diverse, capturing multiple modalities (EHRs, text, graphs, and images), domains (healthcare, content moderation, chemistry), and architectures (spanning logistic regressions to large language models). We report ground truth metrics for the binary and multiclass classifiers in Tables S1 and S2 respectively, and a detailed description of each dataset and classifier set in Sec. B.1. We summarize each dataset and differences between classifiers in the associated classifier set below.

1. **MIMIC-IV** (Johnson et al., 2020): We use risk scores trained to predict three outcomes: (1) critical outcomes, (2) emergency department revisit within 30 days, and (3) hospital admission using MIMIC-IV, a dataset containing health records of patients admitted to the emergency department. For each outcome, we use nine classifiers based on prior work (Movva et al., 2023), which differ by function class and training seed.

2. **CivilComments** (Borkan et al., 2019): We use seven pretrained classifiers provided by the WILDS benchmark, trained on CivilComments (Koh et al., 2021). The classifiers differ in training seed and training loss. Each classifier provides a probabilistic score that each example is flagged as a "toxic" comment by human annotators.

3. **OGB-SARS-CoV** (Hu et al., 2020): We use eight classifiers provided by the WILDS benchmark on the OGB-MolPCBA dataset which differ in training seed and training loss. We focus on the task of predicting whether a molecule inhibits the maturation of a virus (SARS-CoV), as it is the property with the fewest missing labels and a reasonable positive prevalence.

4. **MultiNLI** (Williams et al., 2018): We use four classifiers from the SubpopBench benchmark (Yang et al., 2023) which differ in loss functions and training procedures. Each classifier predicts whether a test sentence is an entailment, contradiction, or neutral.

5. **ImageNetBG** (Xiao et al.): We use four classifiers from the SubpopBench benchmark (Yang et al., 2023) which differ in loss functions and training procedures. Each classifier predicts an image as belonging to one of nine coarse-grained classes.

6. **AG News** (Zhang et al., 2015): We fine-tune ten open-sourced LLMs (sentence transformers) available on HuggingFace for news article classification. The resulting classifiers differ in their base architectures and original training datasets. Once fine-tuned, each classifier predicts news article headlines as belonging to one of four genres.

## 5.2 BASELINES

We compare against ~~five~~seven baselines that (with the exception of *Labeled*) make use of both labeled and unlabeled data to arrive at performance estimates:

1. *Labeled* represents the standard approach to classifiers evaluation and compares classifier scores to ground truth labels, only using examples for which labels are available.

2. *Pseudo-Labeled* trains a logistic regression classifier to predict the true label from the classifier scores—directly estimating $P(y^{(i)}|\mathbf{s}^{(i)})$—and then labels the unlabeled examples using this classifier.

3. *Majority-Vote* ensembles the classifier predictions by performing an accuracy-weighted aggregation of each classifier's prediction. We assign weights to each classifier based on the accuracy achieved on the small sample of labeled data.

4. *Dawid-Skene* (Dawid & Skene, 1979), uses multiple noisy discrete annotations to estimate the latent true label of each example. Dawid-Skene assumes class-conditional errors are independent across annotators (i.e. $P(\hat{y} \neq y|y = k)$ is independent).

5. *Bayesian Calibration* (Ji et al., 2020) re-calibrates a classifier's predicted probabilities based on the observation that if a classifier's scores are calibrated, estimating performance from the predictions alone is possible. Bayesian Calibration does not extend to multiclass settings, so we can only use this baseline in our binary experiments.

6. *AutoEval* (Boyeau et al., 2024) learns a *rectifier* on limited labeled data to debias classifier predictions. This rectifier is then applied to the unlabeled examples to estimate metrics of interest. We compare to AutoEval on accuracy; additional metrics are not directly supported by the public implementation.

7. *Active-Testing* (Kossen et al., 2021) reframes semi-supervised evaluation as an active learning problem, in which a method selects which examples to label out of a large pool of unlabeled examples. The performance metric is then computed according to the sample of labeled data. We compare to Active-Testing on accuracy, a metric for which an acquisition strategy is available.

For baseline implementation details, please refer to Appendix B.2. We also provide a comparison to ensembling (i.e., drawing the "ground truth" label based on the average classifier score) in Appendix D.4, and a discussion of connections to weak supervision in Appendix D.3.

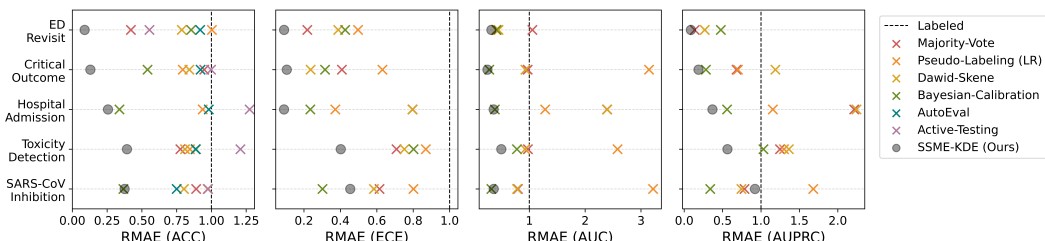

Figure 2: **Metric estimation error on binary tasks ($n_\ell = 20$, $n_u = 1000$).** Each point plots the rescaled mean absolute error (RMAE) across runs, where 1.0 (dashed line) is the RMAE of using labeled data alone. SSME (gray) achieves lower estimation error than do the baselines (averaging across metrics and 50 runs) and reported across 5 binary tasks (y-axis). Tables reporting absolute performance and standard deviations are in Tables 1, S3, S4, and S5.

### 5.3 EVALUATION

We evaluate SSME's ability to estimate four continuous performance metrics for each binary classifier: accuracy, area under the receiver operating characteristic curve (AUC), area under the precision-recall curve (AUPRC), and the expected calibration error (ECE). For multi-class problems, we evaluate accuracy and top-label calibration error (Gupta & Ramdas, 2022); evaluations of multiclass AUC and AUPRC are also possible, but less standard.

We partition each dataset into three splits: the classifier training split (which we use to train the classifiers whose performance we will estimate), the estimation split (which we use to fit SSME and estimate classifier performance), and the evaluation split (which we use for a held-out, ground-truth measure of classifier performance). All splits are sampled from the same distribution, except when estimating subgroup-specific performance, where the evaluation split pertains to a single subset of the test distribution.

To evaluate metric estimates, we measure the absolute error of the *estimated* metric, computed using the estimation split, compared to the *true* metric, computed on the held-out evaluation split (averaging over classifiers in the set). The estimation split consists of either 20, 50, or 100 labeled examples and 1000 unlabeled examples across all experiments. The size of the evaluation split is on the order of thousands of labeled examples and varies by task (see Appendix B.1 for exact split sizes). For each task, we report results over 50 random samples of the splits. In line with prior work, we report rescaled estimation error for each metric (where all errors are relative to using labeled data alone), allowing us to standardize the scale of errors across datasets and metrics (Garg et al., 2022).

## 6 RESULTS

### 6.1 CLASSIFIER EVALUATION ON BINARY TASKS

We now compare SSME to five baselines in terms of its ability to estimate classifier performance on five binary tasks. All figures report rescaled metric estimation error (RMAE; lower is better) and reflect performance estimation using 20 labeled examples and 1000 unlabeled examples. Rescaling metric estimation error allows us to aggregate performance across tasks and metrics, and is standard in prior work (Garg et al., 2022). Results are consistent across additional values of $n_\ell$ (50 and 100; see results in D.1), although labeled data grows more competitive (as expected) as the labeled dataset size increases.

**Comparison to baselines** SSME achieves lower mean estimation error (averaging across tasks and metrics) than all baselines, indicating more accurate estimation of classifier performance. Concretely, SSME reduces estimation error by 5.1× relative to labeled data alone (averaged across tasks and metrics). In contrast, the next best method reduces estimation error by 2.4×. SSME also outperforms baselines on specific metrics. For accuracy, SSME reduces metric estimation error, relative to using labeled data alone, by 5.6× (averaged across tasks); the next best method for each dataset reduces metric estimation error by 2.0×. While the magnitude by which SSME beats baselines varies

| Dataset | $n_\ell$ | $n_u$ | Labeled | Majority-Vote | Pseudo-Labeled | Dawid-Skene | AutoEval | Active-Testing | Bayesian-Calibration | SSME-KDE-M | SSME-KDE (Ours) |
|---|---|---|---|---|---|---|---|---|---|---|---|
| Critical Outcome | 20 | 1000 | 5.19 ± 3.85 | 4.92 ± 0.21 | 4.12 ± 3.87 | 4.36 ± 0.31 | 4.78 ± 3.34 | 5.17 ± 1.16 | 2.80 ± 2.23 | 1.70 ± 0.99 | **0.67 ± 0.46** |
| | 50 | 1000 | 2.90 ± 2.13 | 4.71 ± 0.23 | 3.06 ± 2.32 | 4.07 ± 0.40 | 3.01 ± 2.36 | 5.61 ± 1.44 | 2.07 ± 1.29 | 1.65 ± 0.90 | **0.78 ± 0.47** |
| | 100 | 1000 | 2.09 ± 1.47 | 4.55 ± 0.31 | 1.58 ± 1.08 | 3.87 ± 0.38 | 2.00 ± 1.16 | 5.48 ± 1.25 | 1.18 ± 0.74 | 1.30 ± 0.70 | **0.77 ± 0.47** |
| ED Revisit | 20 | 1000 | 5.11 ± 3.53 | 2.15 ± 0.08 | 5.13 ± 3.23 | 4.02 ± 2.83 | 4.70 ± 3.32 | 2.83 ± 0.71 | 4.36 ± 2.76 | 1.64 ± 1.24 | **0.45 ± 0.36** |
| | 50 | 1000 | 2.02 ± 2.08 | 2.10 ± 0.11 | 2.73 ± 2.24 | 2.74 ± 2.22 | 1.95 ± 2.07 | 3.03 ± 0.96 | 2.47 ± 2.07 | 1.46 ± 0.97 | **0.53 ± 0.39** |
| | 100 | 1000 | 1.43 ± 1.15 | 2.01 ± 0.15 | 1.54 ± 1.22 | 1.51 ± 1.18 | 1.42 ± 1.04 | 2.64 ± 0.57 | 1.43 ± 1.12 | 1.18 ± 0.89 | **0.57 ± 0.39** |
| Hospital Admission | 20 | 1000 | 7.32 ± 4.52 | 19.68 ± 0.44 | 6.86 ± 4.31 | 19.55 ± 0.47 | 7.19 ± 3.73 | 9.33 ± 3.03 | 2.48 ± 1.59 | 3.29 ± 1.71 | **1.88 ± 1.04** |
| | 50 | 1000 | 5.40 ± 2.98 | 18.91 ± 0.51 | 3.99 ± 2.97 | 18.78 ± 0.51 | 5.23 ± 2.46 | 9.25 ± 2.32 | 2.14 ± 1.28 | 3.17 ± 1.85 | **1.95 ± 0.99** |
| | 100 | 1000 | 3.64 ± 1.99 | 18.02 ± 0.52 | 3.01 ± 1.92 | 17.81 ± 0.59 | 4.01 ± 1.99 | 9.24 ± 2.90 | 2.42 ± 1.19 | 3.06 ± 1.64 | **1.51 ± 0.82** |
| SARS-CoV Inhibition | 20 | 1000 | 6.11 ± 3.45 | 5.44 ± 0.21 | 5.95 ± 3.62 | 4.91 ± 0.65 | 4.59 ± 3.05 | 5.97 ± 1.69 | **2.25 ± 1.13** | 3.06 ± 0.83 | 2.30 ± 0.56 |
| | 50 | 1000 | 3.22 ± 2.05 | 5.33 ± 0.23 | 2.99 ± 1.64 | 4.50 ± 0.63 | 2.64 ± 1.53 | 5.74 ± 1.14 | **1.74 ± 0.76** | 2.59 ± 0.94 | 2.35 ± 0.35 |
| | 100 | 1000 | 2.04 ± 1.38 | 5.07 ± 0.27 | 2.14 ± 1.10 | 4.01 ± 0.62 | 1.94 ± 0.90 | 5.99 ± 1.70 | **1.43 ± 0.68** | 1.84 ± 0.85 | 2.36 ± 0.47 |
| Toxicity Detection | 20 | 1000 | 5.95 ± 2.64 | 4.62 ± 0.31 | 5.03 ± 2.91 | 4.82 ± 0.32 | 5.27 ± 2.71 | 7.19 ± 1.57 | 5.29 ± 1.06 | 6.71 ± 0.83 | **2.34 ± 0.52** |
| | 50 | 1000 | 4.03 ± 2.44 | 4.47 ± 0.29 | 2.88 ± 1.72 | 4.65 ± 0.29 | 3.37 ± 1.48 | 7.26 ± 1.71 | 4.57 ± 1.07 | 5.38 ± 1.01 | **2.22 ± 0.47** |
| | 100 | 1000 | 2.43 ± 1.48 | 4.26 ± 0.40 | **1.90 ± 1.11** | 4.46 ± 0.40 | 2.34 ± 0.94 | 7.50 ± 2.16 | 3.78 ± 0.92 | 3.80 ± 1.16 | 2.14 ± 0.54 |

Table 1: **Mean absolute error in accuracy estimation on binary tasks.** We report mean absolute error (averaging across classifiers) across five binary classification tasks and different amounts of labeled data. We bold the best performing method in each row, and underline the next best performing method. SSME-KDE-M describes performance when fitting SSME to a single classifier's scores, instead of modeling the joint distribution of $p(y, s)$.

—for example, SSME reduces error by $2.9\times$ on AUC (averaged across tasks), while the next best method reduces error by $2.6\times$ — SSME consistently outperforms baselines across metrics.

Our results are also encouraging in absolute terms. With 20 labeled examples and 1000 unlabeled examples, SSME estimates accuracy within 1.5 percentage points (averaging across tasks). The closest baseline estimates accuracy within 3.4 percentage points. Results comparing SSME to the next best baseline on other metrics (1.9 vs 3.8 on ECE; 3.6 vs 4.3 on AUC; 8.5 vs 10.2 on AUPRC) confirm that SSME not only achieves more accurate classifier performance estimation compared to prior work, but that the resulting performance estimates are reasonably close to the ground truth measurements for each metric.

SSME estimates performance more accurately because it makes use of all available information: multiple classifiers, continuous scores over all classes, and unlabeled data. *Dawid-Skene*[1] and *Auto-Eval* discretize classifier scores (although *AutoEval* does make use of classifier confidence associated with each discrete prediction). *Pseudo-Labeling*, *Bayesian-Calibration*, and *AutoEval* each learn a mapping from $\mathbf{s}$ to $y$ using only the labeled data and apply that mapping to the unlabeled data (rather than learning from labeled and unlabeled data together). By jointly learning across both labeled and unlabeled data, SSME is able to generalize much better in cases where there aren't enough labels to estimate the joint distribution of classifier scores and labels from labeled examples alone. Finally, *Bayesian-Calibration* learns from a single classifier's scores, and does not learn from multiple models at once.

**Comparison across metrics** SSME provides the greatest benefits relative to labeled data alone when measuring expected calibration error (ECE), with a reduction in estimation error of $7.2\times$ (averaging across tasks). ECE is harder to estimate with few labeled examples because it requires binning and then averaging calibration error across bins. This process tends to yield greater variability when the number of labeled points per bin is small. We observe the smallest benefits relative to labeled data alone when measuring AUPRC (a reduction in estimation error of $2.2\times$, relative to labeled data).

**Comparison across amounts of labeled data** SSME's performance continues to improve with more labeled data, but the advantage it confers over labeled data decreases: for example, with 20, 50, and 100 datapoints, SSME outperforms labeled data alone by $5.6\times$, $3.0\times$, and $1.6\times$. Similar to labeled data alone, there are diminishing but positive returns to adding labeled data to SSME's performance estimation procedure.

Another way to quantify SSME's benefit is to measure the amount of labeled data required to match SSME's performance, or the *effective sample size* (ESS), as introduced by prior work (Boyeau et al., 2024) (see Appendix B.4 for implementation details). With access to 20 labeled exampels and 1000 unlabeled examples, SSME achieves an average ESS of 539 labeled examples for estimating ECE (averaging over tasks). In contrast, the next best approach achieves an ESS of 110 labeled examples to estimate ECE.

---

[1]Fig. 2 omits Dawid-Skene from accuracy results on hospital admission because its much higher RMAE of 2.6 distorts the plotting scale.

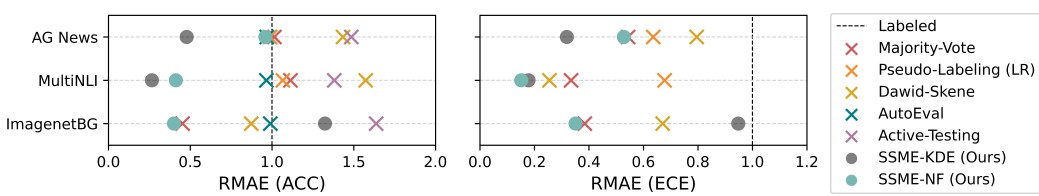

Figure 3: **Metric estimation error on multiclass tasks ($n_\ell = 20$, $n_u = 1000$).** SSME-NF consistently reduces estimation error relative to labeled data alone, and more consistently than any baseline that is able to estimate performance in multiclass settings.

**Comparison to marginal fit**    To validate the benefit of fitting the mixture model to multiple classifiers simultaneously, we compare to an ablated version of SSME fit on a single classifier at a time. In this setting, SSME estimates the classifier-specific marginal distribution of $P(y|\mathbf{s})$, and uses this estimate to evaluate the classifier in question. Doing so results in worse performance estimates across metrics, tasks, and amounts of labeled data (Tables 1, S3, S4, S5) relative to our full model. This agrees with findings from the ensembling literature (Schapire, 1990): each classifier provides distinct information about the ground truth label for a given example, which SSME is able to use.

## 6.2 CLASSIFIER EVALUATION ON MULTI-CLASS TASKS

We now validate SSME's performance on multi-class tasks; Figure 3 reports our results. Because the utility of kernel density estimators is known to degrade in higher dimensional problems (such as here with multi-class outputs) (Jiang, 2017), we provide an instantiation of SSME using a normalizing flow, which we term SSME-NF (for additional implementation details, see Appendix C). Normalizing flows have been shown to effectively model mixtures of high-dimensional distributions (Izmailov et al., 2020); our results align with these findings. SSME-NF improves over labeled data by $2.9\times$ (averaging over tasks and metrics); SSME-KDE improves over labeled data by $2.7\times$. In contrast, the next best baseline (Dawid-Skene) improves over labeled data by $1.5\times$. On each dataset and metric, SSME-NF is consistently one of the top two methods.

**Applications to LLM evaluation**    Our results are encouraging in the context of evaluating large language models as annotators, an emerging application (Ziems et al., 2023). The classifier sets we evaluate on MultiNLI and AG News reflect real multi-class settings where a practitioner might wish to evaluate several LLM-based classifiers: the MultiNLI classifier set contains off-the-shelf LLM-based classifiers, while the AG News classifier set contains the top 10 most-downloaded sentence encoders on HuggingFace, fine-tuned on 200 labeled examples using SetFit (Tunstall et al., 2022).

Our results demonstrate that SSME is able to accurately evaluate LLMs on each task with far fewer examples; with just 20 labeled samples and 1000 unlabeled samples, SSME-NF is able to achieve an effective sample size of 168 for estimating ECE and 103 for estimating accuracy (averaging over tasks). The next best method achieves an effective sample size of 83 for ECE and 20 for accuracy.

## 6.3 ASSESSING SUBGROUP-SPECIFIC PERFORMANCE

SSME can be applied to measure performance within demographic groups of interest, a task central to assessments of algorithmic fairness (Chen et al., 2021a). These groups may be based on gender, age, race, or other groups who face disparities. Unlabeled data has been used in multiple ways to improve subgroup evaluation and algorithmic fairness (Ji et al., 2020; Sagawa et al., 2021; Ktena et al., 2024; Movva et al., 2024): in particular, it can be used to estimate gaps in performance between groups, such as disparities in accuracy across race (Ji et al., 2020).

We conduct our analysis in the context of critical outcome prediction on MIMIC-IV, as prior studies have established that predictive models often display disparities in error rates for this task (Movva et al., 2023). The first two steps involved in SSME remain the same: we (1) acquire classifier scores on all patients using each classifier, and (2) fit the mixture model to classifier scores over the entire sample of labeled and unlabeled data. We then produce subgroup-specific performance estimates by using the empirical distribution of $\mathbf{s}$ *within* each subgroup; that is, we sample ground truth labels $y$

according to our estimated $p(y|\mathbf{s})$ for only those $\mathbf{s}$ observed among, for example, female patients. We then compare our estimated subgroup metric to the the ground truth metric evaluated on a large held-out sample for the given subgroup. We perform this analysis with respect to three categories of demographic groups: age (binned into 10 deciles), sex (male or non-male), and race/ethnicity (White, Black, Asian, or Hispanic/Latino). When there is no labeled data for a given subgroup, *Labeled* estimates subgroup-specific performance as global performance.

We report each method's reduction in estimation error relative to labeled data (averaging over metrics and subgroups) for each demographic category in Table 2, comparing to all baselines which can estimate the four performance metrics we average over (accuracy, ECE, AUC, and AUPRC). SSME reduces metric estimation error by $5.3\times$ on sex, $2.6\times$ on race, and $2.4\times$ on age relative to labeled data, and to a greater extent than all baselines. SSME also outperforms all baselines on all individual metrics except for AUC, for which Bayesian-Calibration reduces estimation error by $3.2\times$ as compared to $2.2\times$ for SSME. Bayesian-Calibration is particularly well-suited to esti-

|  | Age | Sex | Race |
|---|---|---|---|
| Pseudo-Labeling (LR) | 0.92 | 1.06 | 1.13 |
| Dawid-Skene | 0.75 | 0.67 | 0.67 |
| Bayesian-Calibration | 0.45 | 0.33 | 0.41 |
| SSME-KDE (Ours) | **0.42** | **0.19** | **0.39** |

Table 2: **Subgroup-specific performance estimation ($n_l = 20$, $n_u = 1000$).** SSME achieves the lowest rescaled metric estimation error (RMAE) (averaging across metrics and demographic subgroups).

mating AUC because it assumes monotonicity when mapping $s$ to $y$; i.e., $P(y = 1|s^{(i)}) \geq P(y = 1|s^{(j)})$ when $s^{(i)} \geq s^{(j)}$. When the classifiers in question have high AUCs — as the critical outcome classifiers do — this is a useful assumption to make.

### 6.4 Impact of classifier set characteristics on SSME's performance

We now characterize SSME's performance in a semi-synthetic setting, in which we can assess how characteristics like classifier accuracy and calibration affect the performance of the proposed approach. To do so, we create sets of three classifiers based on the widely-used Adult dataset (Becker & Kohavi, 1996), where the task is to predict whether a person's income is above $50K. To create differences between the three classifiers in a set, we train them on random fixed-size samples of 100 labeled examples from different portions of the dataset, partitioned based on age. In doing so, our semi-synthetic classifier sets mimic how training data for different real-world classifiers can differ in demographically meaningful ways. We repeat this procedure to produce 500 sets of three classifiers, where sets differ in the training data provided to each classifier. Our procedure naturally produces random variation in classifier properties, like accuracy and calibration, which we can use to study how well SSME performs. For additional experimental details, refer to Appendix B.3.

**Classifier accuracy** We find that as the average accuracy among classifiers in the set increases, SSME's performance estimation error decreases (Fig. 4, left). This trend holds across each of the four metrics we consider (see Figures S1 and S2 for additional plots) and can be attributed to how more accurate classifiers produce more separable components of the mixture model we aim to estimate. More accurate classifiers allow SSME to better estimate the ground truth label $y$ for unlabeled examples. Classifier accuracy directly impacts the separation of mixture model components, which existing work has shown to result in more accurate parameter estimation (Redner & Walker, 1984).

**Classifier calibration** Similarly, when the classifier set grows more calibrated on average, SSME's estimation error decreases. We can attribute this behavior to how poorly calibrated classifiers result in worse initialization for the mixture model, because we initialize component assignments for the unlabeled examples based on the mean prediction across classifiers.

**Classifier quantity** Conditional on each of the classifier set characteristics discussed (accuracy and calibration), increasing the number of classifiers reduces performance estimation error. Our results suggest that increasing the number of classifiers can sometimes be more beneficial than acquiring a set of more accurate or better calibrated classifiers. Given the widespread availability of pretrained classifiers, this is a promising path towards more accurate evaluations.

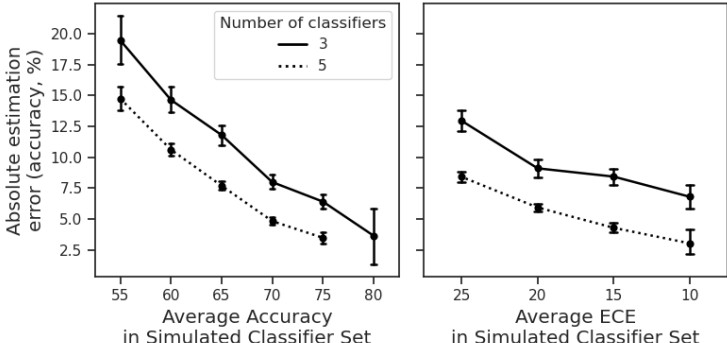

Figure 4: **Impact of classifier set characteristics ($n_\ell = 20$, $n_u = 1000$).** We plot SSME's performance as a function of average classifier set accuracy (left) and average classifier set calibration (right) among a set of semi-synthetic classifier sets, grouped into equal-width bins. More accurate and better calibrated classifier sets produce more accurate performance estimates, as expected. Adding classifiers to the set (dashed) can improve performance estimation to a greater extent than improving the average accuracy of the classifiers in the set: for example, adding two more classifiers typically produces a greater improvement than increasing average classifier accuracy by 5%.

## 7 DISCUSSION

In this paper, we presented Semi-Supervised Model Evaluation (SSME), a method which supplements sparse labeled data with *unlabeled data* to more accurately estimate classifier performance. SSME exploits three aspects of the current machine learning landscape: (i) there are frequently multiple classifiers for the same task, (ii) continuous classifier scores are often available for all classes, and (iii) unlabeled data is often far more plentiful than labeled data. We show that across multiple tasks, architectures, and modalities SSME substantially outperforms using labeled data alone and standard baselines.

These results suggest several directions for future work. First, each of the metrics we examined center on evaluating a single classifier. But because SSME estimates the full joint distribution $P(y, \mathbf{s})$, it could also be used to to measure properties of the classifiers as a set. For instance, recent work has highlighted the importance of measuring *systemic failures* (Kleinberg & Raghavan, 2021) where all classifiers produce errors on the same instances. Second, our experiments assess settings in which the unlabeled data is sampled from the same distribution as the labeled data. Although this is common — for example, when a random subset of examples is annotated — there are other settings where the available unlabeled data systematically differs from the labeled data (Sagawa et al., 2021). Applying SSME to those settings represents a natural direction for future work. Finally, future work could also extend SSME beyond classification to estimate the joint distribution $P(y, \mathbf{s})$ for *continuous $y$* (e.g, using a mixture density network) or *structured $y$* (e.g. graphs) (Vishwakarma & Sala, 2022). More generally, our results strongly indicate that when large amounts of labeled data are unavailable, semi-supervised evaluation can be valuable. SSME is one way to do this, but other approaches are worth exploring.

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

APPENDIX

# A    RELATED WORK

We detail related work in unsupervised performance estimation here. Works below assume access to *only* unlabeled data; in contrast, SSME learns from both labeled and unlabeled data.

**Unsupervised performance estimation**    involves estimating the performance of a model given only unlabeled data. Methods designed to address this problem often focus on out-of-distribution samples, where labeled data is scarce and model performance is known to degrade. Several works have illustrated strong empirical relationships between out-of-distribution generalization and thresholded classifier confidence (Garg et al., 2022), dataset characteristics (Deng & Zheng, 2021; Guillory et al., 2021), in-distribution classifier accuracy (Miller et al., 2021), and classifier agreement (Parisi et al., 2014; Platanios et al., 2017; Baek et al., 2022).

Several works have formalized when unsupervised model evaluation is possible (Donmez et al., 2010; Chen et al., 2022; Garg et al., 2022; Lu et al., 2023), and propose assumptions under which estimates of performance are recoverable. Donmez et al. (2010) and Balasubramanian et al. (2011) assume knowledge of $p(y)$ in the unlabeled sample. Steinhardt & Liang (2016) assume conditionally-independent subsets of the observed features, inspired by conditional-independence assumptions made in works such as Dawid & Skene (1979). Guillory et al. (2021) assume classifier calibration on unlabeled samples. Chen et al. (2022) assume a sparse covariate shift model, in which a subset of the features' class-conditional distribution remains constant. Lu et al. (2023) illustrate misestimation of $p(y)$ in the unlabeled example, and assume that $p(y)$ out-of-distribution is close to $p(y)$ in-distribution. As Garg et al. (2022) highlight, assumptions are necessary to make any claim about the nature of unsupervised model evaluation, and the above methods are a representative sample of assumptions made by prior works. Finally, there has been a surge of interest in unsupervised performance estimation in the context of large language models (Zheng et al., 2023; Huang et al., 2024). A standard approach here is to use a large language model to adjudicate the quality of text generated by other language models. Methods in this literature are often specific to large language models, while SSME is not.

Our work is also similar, in spirit, to methods that learn to debias classifier predictions on a small set of labeled data and then apply that debiasing procedure to classifier predictions on unlabeled examples. Prediction-powered inference (Angelopoulos et al., 2023) and double machine learning (Chernozhukov et al., 2018) both learn a debiasing procedure to ensure that unlabeled metric estimates (e.g., accuracy) are statistically unbiased. One of the baselines we compare to, AutoEval (Boyeau et al., 2024), is built atop prediction-powered inference.

# B    EXPERIMENTAL DETAILS

## B.1    REAL DATASETS AND CLASSIFIER SETS

We provide additional detail for the six datasets we use in our work, including ground truth $p(y)$ for each dataset and ground truth metrics for each classifier in the associated classifier set in Table S1 and Table S2. As discussed, each dataset is split into a training split (provided to each classifier as training data), an estimation split (provided to each performance estimation method), and an evaluation split (used to compute ground truth metrics for each classifier). We determine training splits based on prior work. We then split the remaining data in half (randomly, for each run) to produce the estimation and evaluation splits. We then subsample the estimation split to have $n_l$ labeled examples and $n_u$ unlabeled examples. We ensure that the labeled data always includes at least one example from each class. Thus, the estimation split contains $n_l + n_u$ examples in each experiment, and the evaluation split for each task is fixed across runs (exact sample sizes reported below).

1. **MIMIC-IV**: We use three binary classification tasks from MIMIC-IV (Johnson et al., 2020), a large dataset of electronic health records describing 418K patient visits to an emergency department. We focus on three tasks: **hospitalization** (predicting hospital admission

based on features available during triage, $p(y = 1) = 0.45$), **critical outcomes** (predicting inpatient mortality or a transfer to the ICU within 12 hours, $p(y = 1) = 0.06$), and **emergency department revisits** (predicting a patient's return to the emergency department within 3 days, $p(y = 1) = 0.03$). We split and preprocess data according to prior work (Xie et al., 2022; Movva et al., 2023). No patient appears in more than one split. For each task, the evaluation split contains 70,439 examples. The classifiers in the associated set differ by function class (logistic regression, decision tree, and multi-layer perceptron) and random seed (0, 1, 2).

2. **Toxicity detection**: The task is to predict presence of toxicity given an online comment, using data from CivilComments (Borkan et al., 2019; Koh et al., 2021) where $p(y = 1) = 0.11$. The evaluation split contains 66,891 examples. The classifiers in the associated set differ by training loss (ERM, IRM, and CORAL) and random seed (0, 1, 2).

3. **Biochemical property prediction** The task is to predict presence of a biochemical property based on a molecular graph, using data from the Open Graph Benchmark (Hu et al., 2020). We focus on the task of predicting whether a molecule inhibits SARS-CoV virus maturation, where $p(y = 1) = 0.09$. We filter out examples for which *no* label is observed (i.e. the molecule was not screened at all) because it is impossible to evaluate our performance estimates on those examples. Doing so reduces data held-out from training from 43,793 to 28,325 examples. The evaluation split then contains half, or 14,163, of those examples. The classifiers in the associated set differ by training loss (ERM, IRM, and CORAL) and random seed (0, 1, 2).

4. **News classification** The task is to predict one of four news types based on the title and description of an article (Zhang et al., 2015). The classes are balanced and the evaluation split contains 3,800 examples. Classifiers differ by the base LLM fine-tuned to perform news classification. We fine-tune each LLM by training a classification head atop the embeddings from each LLM using the training split provided by HuggingFace and use the classifier probabilities as scores **s** for SSME.

5. **Sentence classification** The task is to predict one of three textual entailments from a sentence (Williams et al., 2018). The classes are balanced and the evaluation split contains 61,856 examples. Classifiers differ by training loss (ReWeight, ReSample, IRM, and SqrtReWeight) according to (Yang et al., 2023).

6. **Image classification** The task is to predict one of nine coarse image categories (e.g. "dog" or "vehicle") from an image (Xiao et al.). The classes are balanced and the evaluation split contains 2,025 examples. Classifiers differ by training loss (ReWeight, ReSample, IRM, and SqrtReWeight) according to (Yang et al., 2023).

## B.2 BASELINES

For baselines that require discrete predictions (i.e. Dawid-Skene and AutoEval), we discretize classifier scores by assigning a class according to the maximum classifier score across classes. We expand on our implementation of each baseline below.

- *Labeled*: When estimating performance over the whole dataset, we compare the classifier scores to the ground truth labels within the labeled sample. However, when estimating subgroup-specific performance, it is often the case that there are no labeled examples for a given subgroup. In these instances, *Labeled* reverts to estimating subgroup-specific performance as performance over all labeled examples.

- *Pseudo-Labeling*: We train a logistic regression with the default parameters associated with the scikit-learn implementation (Pedregosa et al., 2011). Experiments with alternative function classes (e.g. a KNN) revealed no significant differences in performance.

- *Bayesian-Calibration*: Bayesian-Calibration operates on each classifier individually. We make use of the implementation made available by Ji et al. (2020). Extending the proposed approach to multi-class tasks is not straightforward, so we compare to *Bayesian-Calibration* only on binary tasks.

- *Dawid-Skene*: We implement Dawid-Skene with a tolerance of 1e-5 and a maximum number of EM iterations of 100 (the default parameters), using the following public implementation: https://github.com/dallascard/dawid_skene. Dawid-Skene

| Dataset | Classifier | Acc | ECE | AUC | AUPRC |
|---|---|---|---|---|---|
| Hospital Admission | DT-RandomForest-seed1 | 74.2 | 1.5 | 81.5 | 76.0 |
| | MLP-ERM-seed2 | 74.4 | 1.4 | 81.7 | 76.7 |
| | MLP-ERM-seed1 | 74.4 | 1.9 | 81.9 | 77.0 |
| | MLP-ERM-seed0 | 74.5 | 2.4 | 82.0 | 77.0 |
| | LR-LBFGS-seed2 | 73.3 | 4.0 | 80.7 | 75.5 |
| | LR-LBFGS-seed1 | 73.3 | 4.0 | 80.7 | 75.5 |
| | LR-LBFGS-seed0 | 73.4 | 2.9 | 81.0 | 75.7 |
| | DT-RandomForest-seed2 | 74.3 | 1.6 | 81.5 | 76.1 |
| | DT-RandomForest-seed0 | 74.1 | 1.5 | 81.5 | 76.1 |
| Critical Outcome | MLP-ERM-seed2 | 93.9 | 0.9 | 87.9 | 38.6 |
| | MLP-ERM-seed1 | 93.9 | 0.8 | 88.1 | 39.0 |
| | LR-LBFGS-seed2 | 93.6 | 1.2 | 87.6 | 34.2 |
| | MLP-ERM-seed0 | 93.9 | 0.5 | 87.5 | 37.8 |
| | LR-LBFGS-seed0 | 93.6 | 1.2 | 87.6 | 34.1 |
| | DT-RandomForest-seed2 | 94.0 | 0.3 | 87.2 | 38.2 |
| | DT-RandomForest-seed1 | 94.0 | 0.4 | 87.4 | 38.3 |
| | DT-RandomForest-seed0 | 94.0 | 0.4 | 87.4 | 38.3 |
| | LR-LBFGS-seed1 | 93.6 | 1.2 | 87.6 | 34.2 |
| ED Revisit | DT-RandomForest-seed0 | 97.7 | 1.8 | 54.9 | 2.7 |
| | DT-RandomForest-seed1 | 97.7 | 1.7 | 55.3 | 2.7 |
| | DT-RandomForest-seed2 | 97.7 | 1.8 | 54.9 | 2.7 |
| | LR-LBFGS-seed0 | 97.7 | 0.4 | 59.3 | 3.0 |
| | LR-LBFGS-seed2 | 97.7 | 0.4 | 59.1 | 3.0 |
| | MLP-ERM-seed0 | 97.7 | 0.3 | 59.8 | 3.1 |
| | MLP-ERM-seed1 | 97.7 | 0.3 | 59.8 | 3.1 |
| | MLP-ERM-seed2 | 97.7 | 0.5 | 57.9 | 3.0 |
| | LR-LBFGS-seed1 | 97.7 | 0.4 | 59.1 | 3.0 |
| Toxicity Detection | distilbert-CORAL-seed0 | 88.3 | 6.0 | 86.2 | 40.0 |
| | distilbert-IRM-seed2 | 88.7 | 10.2 | 91.9 | 65.5 |
| | distilbert-IRM-seed1 | 89.0 | 9.8 | 91.0 | 66.5 |
| | distilbert-IRM-seed0 | 88.1 | 10.6 | 91.6 | 65.9 |
| | distilbert-ERM-seed2 | 92.1 | 4.9 | 94.1 | 73.3 |
| | distilbert-ERM-seed1 | 92.2 | 6.2 | 93.8 | 72.3 |
| | distilbert-ERM-seed0 | 92.2 | 6.1 | 93.8 | 72.2 |
| Molecule Property 60 | gin-virtual-CORAL-seed1 | 92.8 | 5.2 | 90.1 | 61.9 |
| | gin-virtual-CORAL-seed2 | 92.8 | 5.2 | 90.1 | 61.9 |
| | gin-virtual-ERM-seed0 | 94.6 | 1.2 | 94.5 | 73.5 |
| | gin-virtual-ERM-seed1 | 92.4 | 5.6 | 90.7 | 61.1 |
| | gin-virtual-ERM-seed2 | 92.8 | 5.2 | 90.1 | 61.9 |
| | gin-virtual-IRM-seed0 | 93.2 | 1.8 | 90.2 | 58.4 |
| | gin-virtual-IRM-seed1 | 91.1 | 5.2 | 83.8 | 43.8 |
| | gin-virtual-IRM-seed2 | 91.1 | 5.7 | 82.8 | 44.7 |

Table S1: **Ground truth classifier metrics on binary tasks.** We report ground truth performance for classifiers in the sets associated with each binary task. Each classifier name begins with the architecture (e.g. DT represents DecisionTree), the loss or training procedure (e.g. ERM or IRM), and then the seed. Note that the equivalent accuracies on ED Revisit are a byproduct of both the low class prevalence and the poor classifiers.

| Dataset | Classifier | Acc | ECE |
|---|---|---|---|
| AG News | all-MiniLM-L12-v2 | 84.8 | 4.2 |
|  | mxbai-embed-large-v1 | 85.0 | 14.4 |
|  | multi-qa-MiniLM-L6-cos-v1 | 85.6 | 5.2 |
|  | bge-small-en-v1.5 | 85.2 | 16.9 |
|  | bge-large-en-v1.5 | 86.8 | 4.8 |
|  | bge-base-en-v1.5 | 86.6 | 5.6 |
|  | all-mpnet-base-v2 | 86.7 | 2.9 |
|  | all-MiniLM-L6-v2 | 83.8 | 3.8 |
|  | paraphrase-multilingual-MiniLM-L12-v2 | 85.1 | 9.6 |
|  | paraphrase-MiniLM-L6-v2 | 86.0 | 8.9 |
| MultiNLI | distilbert-SqrtReWeight | 81.4 | 9.2 |
|  | distilbert-ReWeight | 80.9 | 7.4 |
|  | distilbert-ReSample | 81.4 | 8.2 |
|  | distilbert-IRM | 64.8 | 6.1 |
| ImagenetBG | ResNet-ReWeight | 86.6 | 7.8 |
|  | ResNet-ReSample | 87.4 | 7.7 |
|  | ResNet-Mixup | 88.6 | 7.7 |
|  | ResNet-IRM | 54.1 | 30.9 |

Table S2: **Ground truth classifier metrics on multiclass tasks.** We report ground truth performance for classifiers in the sets associated with each multiclass task. Each of the LLMs fine-tuned for AG News are sentence transformers, while the MultiNLI classifiers all use DistilBERT (Sanh, 2019) as the base architecture. The base architecture on ImagenetBG is a ResNet-50.

accepts discrete predictions, so we discretize classifier predictions using thresholding the predicted class probability at $\frac{1}{K}$.

- *Majority-Vote*: We implement Majority-Vote as the accuracy-weighted average of discrete predictions made by each classifier. We discretize predictions by thresholding predicted probabilities at $\frac{1}{K}$. We weight each classifier in proportion to its accuracy on the available labeled data.

- *Active-Testing*: We implement Active-Testing, where the method selects a fixed number of examples to label out of a pool of unlabeled examples, according to the approach proposed by Kossen et al.. We select examples according to the acquisition strategy for estimating accuracy, a metric for which a public implementation is available, and limit our comparison to this metric.

- *AutoEval*: We implement AutoEval using an implementation made available by the authors (Boyeau et al., 2024). The implementation, to the best of our knowledge, only supports accuracy estimation across a set of classifiers, so we limit our comparison to this metric.

### B.3 SEMISYNTHETIC DATASET AND CLASSIFIER SETS

As with the real datasets, we produce three splits: a training split to learn the classifiers (50 examples), an estimation split for the performance estimation methods (20 labeled examples and 1000 unlabeled examples), and an evaluation split to measure ground truth values for each metric (10,000 examples). Each classifier is a logistic regression with default L2 regularization.

### B.4 COMPUTING EFFECTIVE SAMPLE SIZE

In order to compute effective sample size, we produce 50 samples of labeled data for each increment of 5 between 10 labeled examples and 1000. We then compute the mean absolute metric estimation error of using labeled data alone, across all runs. The effective sample size of a given semi-supervised evaluation method is thus the amount of labeled data which achieves the most similar mean absolute metric estimation error.

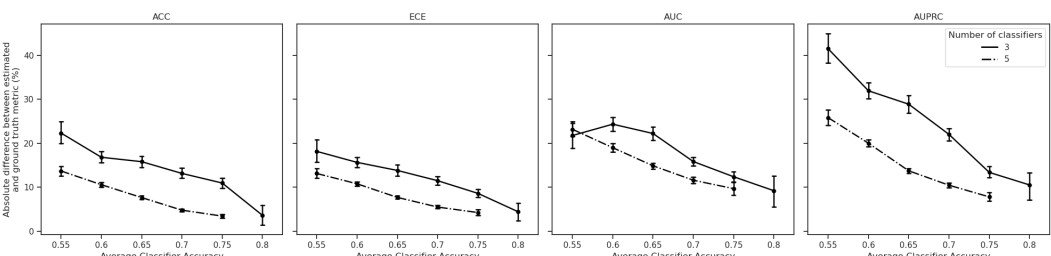

Figure S1: **Impact of average accuracy across classifiers in set on SSME's performance.**

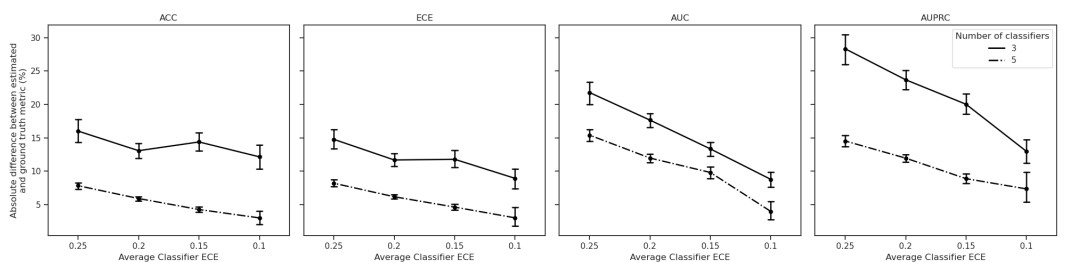

Figure S2: **Impact of average ECE across classifiers in set on SSME's performance.**

## C   NORMALIZING FLOW

One alternative parameterization is to use a normalizing flow to model our mixture of distributions. Normalizing flows learn and apply an invertible transform $f_\theta$ to a random variable $\mathbf{z} \sim D_1$ to obtain $f_\theta(\mathbf{z}) \sim D_2$. Here, we set $\mathbf{z} \sim D_1$ to a Gaussian mixture model and learn a transformation such that $f_\theta(\mathbf{z}) \overset{\text{dist.}}{\approx} \mathbf{s}$, i.e., the transformed distribution roughly matches our classifier score distribution. By modeling $\mathbf{z}$ explicitly as a Gaussian mixture model, one can move back and forth between the two distributions, as $f_\theta^{-1}(f_\theta(\mathbf{z})) = \mathbf{z}$. Specifically, we set the distribution of $\mathbf{Z}$ to follow a Gaussian mixture:

$$\mathbf{Z}|(Y = k) \sim \mathcal{N}(\mu_k, \Sigma_k)$$

Thus, the marginal distribution of $\mathbf{Z}$ is $p_{\mathbf{Z}}(\mathbf{z}) = \sum_{k=1}^{K} \mathcal{N}(\mathbf{z}|\mu_k, \Sigma_k) \cdot p(y = k)$ is the overall density of $\mathbf{z}$. We apply our invertible transformation $f_\theta$ to obtain $\mathbf{s} = f_\theta(\mathbf{z})$. To find $p(\mathbf{s}|y = k)$, we follow the approach of Izmailov et al. (2020):

$$p_{\mathbf{S}}(\mathbf{s}|y = k) = \mathcal{N}(f_\theta^{-1}(\mathbf{s})|\mu_k, \Sigma_k) \cdot \left| \det\left( \frac{\delta f}{\delta x} \right) \right| \cdot p(y = k)$$

Intuitively, we transform $(\mathbf{s}, y)$ into a distribution $(\mathbf{z}, y)$ which follows a Gaussian mixture model. By enforcing the constraint that this transform is invertible, the joint distribution on $(\mathbf{z}, y)$ captures all the information in $(\mathbf{s}, y)$.

We use the RealNVP architecture (Dinh et al., 2016) to parameterize $f_\theta$ using 10 coupling layers, 3 fully-connected layers, and a hidden dimension of 128 between the fully connected layers. Our normalizing flow is lightweight and trains in less than a minute for each dataset in our experiments section using 1 80GB NVIDIA A100 GPU.

Note there are two optimizations here: (1) the normalizing flow transformation $f_\theta$ which maps $\mathbf{s}$ into our latent Gaussian mixture space and (2) the Gaussian mixture model parameters $\mu_k, \Sigma_k$ themselves. We begin by fixing the GMM parameters $\mu_k, \Sigma_k$ to values estimated from our classifier scores $\mathbf{s}$ and learning only the flow $f_\theta$ for 300 epochs. Afterwards, we optimize the GMM parameters $\mu_k, \Sigma_k$ with EM for another 700 epochs.

| Dataset | $n_\ell$ | $n_u$ | Labeled | Majority-Vote | Pseudo-Labeling (LR) | Dawid-Skene | Bayesian-Calibration | SSME-KDE-M | SSME-KDE (Ours) |
|---|---|---|---|---|---|---|---|---|---|
| Critical Outcome | 20 | 1000 | 11.01 ± 4.04 | 4.49 ± 0.31 | 6.94 ± 2.30 | 2.61 ± 0.33 | 3.48 ± 2.76 | 3.17 ± 1.10 | **1.16 ± 0.48** |
| | 50 | 1000 | 6.22 ± 2.23 | 4.30 ± 0.41 | 5.38 ± 1.40 | 2.37 ± 0.32 | 2.56 ± 1.57 | 3.01 ± 0.94 | **1.13 ± 0.47** |
| | 100 | 1000 | 4.20 ± 1.38 | 4.05 ± 0.43 | 3.63 ± 0.77 | 2.25 ± 0.37 | 1.69 ± 0.91 | 2.81 ± 0.81 | **1.15 ± 0.38** |
| ED Revisit | 20 | 1000 | 8.37 ± 3.14 | 1.83 ± 0.07 | 4.16 ± 2.96 | 3.25 ± 2.45 | 3.57 ± 2.78 | 1.88 ± 0.86 | **0.76 ± 0.16** |
| | 50 | 1000 | 4.82 ± 1.73 | 1.78 ± 0.11 | 2.29 ± 1.69 | 2.29 ± 1.68 | 2.04 ± 1.66 | 1.83 ± 0.67 | **0.73 ± 0.18** |
| | 100 | 1000 | 3.29 ± 0.88 | 1.70 ± 0.14 | 1.36 ± 0.78 | 1.34 ± 0.76 | 1.16 ± 0.73 | 1.51 ± 0.63 | **0.73 ± 0.21** |
| Hospital Admission | 20 | 1000 | 21.76 ± 4.18 | 17.34 ± 0.47 | 8.10 ± 4.61 | 17.31 ± 0.42 | 5.12 ± 3.94 | 5.54 ± 1.32 | **1.97 ± 0.47** |
| | 50 | 1000 | 12.74 ± 2.25 | 16.63 ± 0.48 | 5.02 ± 2.45 | 16.60 ± 0.43 | 3.49 ± 2.05 | 5.20 ± 1.19 | **2.06 ± 0.67** |
| | 100 | 1000 | 8.56 ± 1.39 | 15.71 ± 0.46 | 3.91 ± 1.76 | 15.62 ± 0.44 | 3.23 ± 1.68 | 5.32 ± 1.32 | **1.70 ± 0.54** |
| SARS-CoV Inhibition | 20 | 1000 | 7.44 ± 3.44 | 4.57 ± 0.18 | 5.96 ± 3.13 | 4.35 ± 0.53 | **2.24 ± 1.19** | 2.57 ± 0.64 | 3.38 ± 0.47 |
| | 50 | 1000 | 3.66 ± 1.80 | 4.59 ± 0.16 | 3.06 ± 1.28 | 4.08 ± 0.57 | **1.73 ± 0.93** | 2.27 ± 0.72 | 3.41 ± 0.41 |
| | 100 | 1000 | 2.18 ± 1.14 | 4.65 ± 0.19 | 2.36 ± 0.78 | 3.67 ± 0.59 | **1.35 ± 0.78** | 1.79 ± 0.69 | 3.44 ± 0.47 |
| Toxicity Detection | 20 | 1000 | 5.85 ± 2.89 | 4.15 ± 0.32 | 5.09 ± 2.87 | 4.40 ± 0.33 | 4.69 ± 1.21 | 5.67 ± 0.68 | **2.35 ± 0.46** |
| | 50 | 1000 | 3.99 ± 2.28 | 3.99 ± 0.29 | 3.04 ± 1.66 | 4.20 ± 0.26 | 3.97 ± 1.21 | 4.57 ± 0.93 | **2.26 ± 0.44** |
| | 100 | 1000 | 2.37 ± 1.35 | 3.86 ± 0.38 | **1.91 ± 0.99** | 4.10 ± 0.32 | 3.30 ± 0.91 | 3.43 ± 1.05 | 2.19 ± 0.53 |

Table S3: **Mean absolute error in ECE estimation on binary tasks.**

| Dataset | $n_\ell$ | $n_u$ | Labeled | Majority-Vote | Pseudo-Labeling (LR) | Dawid-Skene | Bayesian-Calibration | SSME-KDE-M | SSME-KDE (Ours) |
|---|---|---|---|---|---|---|---|---|---|
| Critical Outcome | 20 | 1000 | 10.09 ± 4.84 | 9.80 ± 1.15 | 31.73 ± 3.95 | 9.39 ± 1.25 | 2.84 ± 0.91 | 4.72 ± 2.27 | **2.52 ± 1.24** |
| | 50 | 1000 | 7.50 ± 4.62 | 8.19 ± 2.12 | 27.33 ± 5.51 | 8.49 ± 1.46 | 3.17 ± 1.17 | 5.61 ± 4.61 | **2.39 ± 1.74** |
| | 100 | 1000 | 5.65 ± 3.44 | 7.31 ± 2.09 | 20.43 ± 4.38 | 7.97 ± 1.08 | **2.70 ± 0.94** | 3.82 ± 1.72 | 2.83 ± 2.89 |
| ED Revisit | 20 | 1000 | 18.48 ± 6.68 | 19.48 ± 8.75 | 7.48 ± 0.72 | 8.27 ± 3.80 | 7.65 ± 0.55 | 11.89 ± 4.66 | **5.92 ± 3.14** |
| | 50 | 1000 | 17.37 ± 7.13 | 17.97 ± 9.26 | 7.48 ± 0.95 | 7.62 ± 0.99 | 7.30 ± 0.76 | 11.99 ± 4.36 | **5.09 ± 2.56** |
| | 100 | 1000 | 14.13 ± 6.03 | 14.84 ± 8.40 | 7.06 ± 1.46 | 7.09 ± 1.52 | 7.47 ± 1.17 | 11.28 ± 5.73 | **5.08 ± 2.77** |
| Hospital Admission | 20 | 1000 | 6.97 ± 4.64 | 16.66 ± 0.30 | 8.94 ± 5.97 | 16.70 ± 0.31 | 2.67 ± 1.15 | 3.63 ± 1.95 | **2.51 ± 1.38** |
| | 50 | 1000 | 5.08 ± 3.49 | 16.09 ± 0.36 | 5.59 ± 4.31 | 16.18 ± 0.31 | 2.62 ± 1.65 | 3.18 ± 1.95 | **2.51 ± 1.20** |
| | 100 | 1000 | 3.57 ± 2.58 | 15.29 ± 0.43 | 3.66 ± 2.68 | 15.32 ± 0.39 | 2.55 ± 1.34 | 3.17 ± 1.60 | **2.02 ± 1.20** |
| SARS-CoV Inhibition | 20 | 1000 | 9.61 ± 9.22 | 7.64 ± 1.21 | 30.92 ± 4.35 | 7.50 ± 1.05 | **3.07 ± 1.00** | 5.42 ± 2.63 | 3.48 ± 1.58 |
| | 50 | 1000 | 5.84 ± 3.64 | 6.86 ± 1.59 | 22.71 ± 4.29 | 7.06 ± 1.04 | 3.62 ± 0.97 | 5.02 ± 1.86 | **3.41 ± 1.68** |
| | 100 | 1000 | 3.97 ± 1.97 | 5.33 ± 1.60 | 16.33 ± 3.27 | 6.04 ± 1.17 | 3.53 ± 1.35 | 4.21 ± 1.90 | **3.46 ± 1.63** |
| Toxicity Detection | 20 | 1000 | 6.71 ± 3.57 | 6.54 ± 0.32 | 17.32 ± 7.51 | 6.20 ± 0.41 | 5.22 ± 0.59 | 6.05 ± 1.02 | **3.34 ± 0.82** |
| | 50 | 1000 | 4.76 ± 3.29 | 6.30 ± 0.28 | 11.79 ± 6.41 | 5.97 ± 0.33 | 4.76 ± 0.74 | 4.86 ± 1.03 | **3.15 ± 0.66** |
| | 100 | 1000 | 3.82 ± 2.17 | 6.14 ± 0.44 | 7.54 ± 3.73 | 5.84 ± 0.44 | 4.25 ± 0.96 | 4.15 ± 1.20 | **3.09 ± 0.81** |

Table S4: **Mean absolute error in AUC estimation on binary tasks.**

# D SUPPLEMENTARY RESULTS

## D.1 RESULTS REPORTING MEAN ABSOLUTE ERROR

In the main text, we evaluate our method and all baselines using 20 labeled examples and 1000 unlabeled examples and report *rescaled* mean absolute error across metrics and tasks. Here, we supplement those results by reporting mean absolute error across each task and metric and expanding $n_l$ to include 50 and 100. The number of unlabeled examples remains the same (1000) to isolate the effect of additional labeled data.

Tables 1, S3, S4, and S5 report our results on each binary task, for accuracy, ECE, AUC, and AUPRC, respectively. Three high-level findings emerge. First, SSME-KDE achieves the lowest mean absolute error (averaging across tasks and amounts of labeled data). Second, SSME-KDE consistently outperforms the ablated version of SSME, fit to a single model at a time (SSME-KDE-M). And finally, SSME-KDE is able to produce performance estimates that are quite close, in absolute terms, to ground truth. For example, when given 20 labeled examples and 1000 unlabeled examples, SSME-KDE estimates accuracy within at most 2.5 percentage points of ground truth accuracy (across tasks).

Tables S6 and S7 report our results on the multiclass tasks, for accuracy and ECE respectively. Note that we exclude Bayesian-Calibration from multiclass comparisons because the method does not natively support multiclass recalibration. We also omit AutoEval from Table S7 because the implementation of expected calibration error within the framework is not straightforward.

## D.2 RESULTS REPORTING ABSOLUTE PERFORMANCE ESTIMATES

Results thus far have reported aggregate errors in performance estimates across classifiers in the set. Here, we include results on a per-classifier basis in the context of toxicity detection, the task for which we have the largest variability in classifier quality (Tables S8, S9, S10, S11). The tables illustrate how SSME's performance manifests on a per-classifier basis, often producing more accurate estimates than the baselines on the lowest performance classifiers. The tables also make evident that

| Dataset | $n_\ell$ | $n_u$ | Labeled | Majority-Vote | Pseudo-Labeling (LR) | Dawid-Skene | Bayesian-Calibration | SSME-KDE-M | SSME-KDE (Ours) |
|---|---|---|---|---|---|---|---|---|---|
| Critical Outcome | 20 | 1000 | 32.86 ± 18.26 | 22.41 ± 8.36 | 22.98 ± 6.69 | 39.02 ± 4.26 | 9.29 ± 6.01 | 11.48 ± 5.46 | **6.11 ± 2.63** |
| | 50 | 1000 | 22.81 ± 13.16 | 17.44 ± 7.60 | 20.48 ± 8.04 | 35.64 ± 5.80 | 9.34 ± 5.17 | 11.98 ± 5.17 | **6.17 ± 3.47** |
| | 100 | 1000 | 15.71 ± 8.81 | 16.22 ± 6.66 | 14.45 ± 7.30 | 33.31 ± 4.33 | 8.96 ± 5.07 | 11.30 ± 6.01 | **5.77 ± 2.80** |
| ED Revisit | 20 | 1000 | 19.18 ± 13.27 | 2.63 ± 0.44 | 5.14 ± 3.20 | 5.12 ± 4.68 | 9.14 ± 3.74 | 5.07 ± 2.89 | **1.67 ± 1.06** |
| | 50 | 1000 | 8.85 ± 8.14 | 2.44 ± 0.34 | 2.72 ± 2.22 | 3.04 ± 2.80 | 6.03 ± 2.95 | 3.79 ± 2.33 | **1.81 ± 1.08** |
| | 100 | 1000 | 6.34 ± 5.57 | 2.46 ± 1.73 | **1.57 ± 1.19** | 1.74 ± 1.24 | 4.23 ± 1.97 | 3.92 ± 2.23 | 1.82 ± 1.13 |
| Hospital Admission | 20 | 1000 | 9.43 ± 5.85 | 20.87 ± 0.51 | 10.89 ± 9.33 | 21.15 ± 0.52 | 5.26 ± 3.84 | 4.36 ± 1.60 | **3.47 ± 2.04** |
| | 50 | 1000 | 7.46 ± 4.74 | 19.96 ± 0.70 | 7.91 ± 5.89 | 20.34 ± 0.59 | 4.43 ± 2.68 | 3.70 ± 2.26 | **3.64 ± 2.16** |
| | 100 | 1000 | 5.51 ± 3.48 | 18.97 ± 0.70 | 4.12 ± 3.66 | 19.17 ± 0.68 | 3.49 ± 2.17 | 4.00 ± 2.19 | **2.80 ± 1.84** |
| SARS-CoV Inhibition | 20 | 1000 | 22.27 ± 10.94 | 17.50 ± 3.17 | 37.41 ± 8.82 | 16.60 ± 3.91 | **7.54 ± 2.74** | 13.81 ± 5.52 | 20.51 ± 6.12 |
| | 50 | 1000 | 15.02 ± 8.77 | 14.94 ± 4.23 | 30.29 ± 9.40 | 15.01 ± 3.85 | **8.40 ± 3.45** | 12.82 ± 3.63 | 21.06 ± 5.46 |
| | 100 | 1000 | 11.53 ± 5.64 | 13.37 ± 2.75 | 20.34 ± 6.46 | 12.61 ± 3.78 | **8.27 ± 3.19** | 11.01 ± 5.02 | 20.67 ± 5.92 |
| Toxicity Detection | 20 | 1000 | 19.34 ± 8.45 | 24.12 ± 1.49 | 25.12 ± 12.69 | 26.34 ± 1.31 | 19.94 ± 4.70 | 23.38 ± 2.64 | **10.89 ± 3.14** |
| | 50 | 1000 | 13.78 ± 6.52 | 23.11 ± 1.43 | 20.15 ± 12.57 | 25.24 ± 1.31 | 16.84 ± 5.68 | 18.90 ± 3.88 | **9.91 ± 3.06** |
| | 100 | 1000 | 10.69 ± 6.16 | 22.34 ± 1.99 | 14.06 ± 7.21 | 24.51 ± 1.68 | 14.15 ± 5.35 | 14.59 ± 4.54 | **9.88 ± 3.51** |

Table S5: **Mean absolute error in AUPRC estimation on binary tasks.**

| Dataset | $n_\ell$ | $n_u$ | Labeled | Majority-Vote | Pseudo-Labeling | Dawid-Skene | AutoEval | Active-Testing | SSME-KDE | SSME-NF |
|---|---|---|---|---|---|---|---|---|---|---|
| AG News | 20 | 1000 | 5.79 ± 3.04 | 5.88 ± 0.68 | 5.72 ± 4.16 | 8.31 ± 0.54 | 5.61 ± 2.77 | 8.60 ± 2.02 | 2.77 ± 0.96 | 5.56 ± 0.75 |
| | 50 | 1000 | 4.09 ± 1.92 | 5.73 ± 0.87 | 2.97 ± 2.00 | 8.06 ± 0.68 | 3.68 ± 1.48 | 8.18 ± 2.28 | 2.72 ± 1.09 | 5.64 ± 1.03 |
| | 100 | 1000 | 2.93 ± 1.52 | 5.52 ± 0.82 | 2.36 ± 1.48 | 7.66 ± 0.69 | 2.70 ± 1.29 | 8.23 ± 1.83 | 2.50 ± 1.09 | 5.32 ± 1.05 |
| ImagenetBG | 20 | 1000 | 6.62 ± 2.74 | 2.99 ± 0.90 | 33.45 ± 2.96 | 5.78 ± 0.71 | 6.55 ± 2.62 | 10.83 ± 5.34 | 8.76 ± 1.00 | 2.65 ± 0.67 |
| | 50 | 1000 | 3.98 ± 1.63 | 3.01 ± 0.61 | 17.88 ± 2.78 | 5.69 ± 0.73 | 3.87 ± 1.56 | 12.25 ± 7.28 | 8.18 ± 0.90 | 2.66 ± 0.81 |
| | 100 | 1000 | 2.97 ± 1.38 | 2.73 ± 0.57 | 9.37 ± 1.53 | 5.34 ± 0.63 | 2.73 ± 1.13 | 9.08 ± 4.22 | 8.02 ± 0.90 | 2.10 ± 0.68 |
| MultiNLI | 20 | 1000 | 7.46 ± 3.88 | 8.30 ± 0.81 | 7.95 ± 4.55 | 11.73 ± 0.55 | 7.20 ± 3.76 | 10.30 ± 4.14 | 1.98 ± 0.88 | 3.08 ± 0.65 |
| | 50 | 1000 | 4.42 ± 1.99 | 8.14 ± 0.62 | 3.08 ± 2.25 | 11.41 ± 0.52 | 4.17 ± 1.96 | 11.85 ± 3.97 | 1.90 ± 0.76 | 2.79 ± 0.81 |
| | 100 | 1000 | 3.27 ± 1.65 | 7.54 ± 0.69 | 2.47 ± 1.86 | 10.72 ± 0.54 | 3.17 ± 1.59 | 11.63 ± 4.14 | 2.02 ± 0.82 | 2.52 ± 0.77 |

Table S6: **Mean absolute error in accuracy estimation on multiclass tasks.**

| Dataset | $n_\ell$ | $n_u$ | Labeled | Majority-Vote | Pseudo-Labeling | Dawid-Skene | SSME-KDE | SSME-NF |
|---|---|---|---|---|---|---|---|---|
| AG News | 20 | 1000 | 7.04 ± 2.22 | 3.83 ± 0.35 | 4.48 ± 3.23 | 5.60 ± 0.28 | 2.24 ± 0.51 | 3.72 ± 0.50 |
| | 50 | 1000 | 4.85 ± 1.54 | 3.75 ± 0.42 | 2.28 ± 1.36 | 5.37 ± 0.34 | 2.24 ± 0.59 | 3.81 ± 0.46 |
| | 100 | 1000 | 3.24 ± 1.15 | 3.53 ± 0.37 | 1.89 ± 0.96 | 5.02 ± 0.40 | 2.15 ± 0.55 | 3.53 ± 0.60 |
| ImagenetBG | 20 | 1000 | 7.10 ± 2.79 | 2.73 ± 0.84 | 29.64 ± 2.84 | 4.76 ± 0.56 | 6.73 ± 0.54 | 2.49 ± 0.60 |
| | 50 | 1000 | 4.00 ± 1.85 | 2.73 ± 0.52 | 14.18 ± 2.51 | 4.68 ± 0.48 | 6.42 ± 0.57 | 2.49 ± 0.72 |
| | 100 | 1000 | 2.75 ± 1.13 | 2.56 ± 0.64 | 6.68 ± 1.04 | 4.54 ± 0.52 | 6.30 ± 0.62 | 1.96 ± 0.60 |
| MultiNLI | 20 | 1000 | 11.57 ± 4.06 | 3.87 ± 0.43 | 7.84 ± 4.12 | 2.95 ± 0.29 | 2.06 ± 0.88 | 1.75 ± 0.62 |
| | 50 | 1000 | 6.14 ± 2.42 | 3.94 ± 0.32 | 3.10 ± 2.24 | 2.92 ± 0.37 | 2.06 ± 0.72 | 1.63 ± 0.59 |
| | 100 | 1000 | 4.52 ± 1.83 | 3.82 ± 0.34 | 2.37 ± 1.66 | 3.18 ± 0.30 | 2.19 ± 0.76 | 1.45 ± 0.57 |

Table S7: **Mean absolute error in ECE estimation on multiclass tasks.**

| model | Labeled | Majority-Vote | Pseudo-Labeled | Dawid-Skene | Bayesian-Calibration | AutoEval | Active-Testing | SSME-KDE (Ours) | Ground Truth |
|---|---|---|---|---|---|---|---|---|---|
| distilbert-CORAL | 83.90 ± 7.16 | 87.68 ± 0.96 | 86.15 ± 4.11 | 84.75 ± 1.35 | 87.82 ± 4.41 | 85.04 ± 5.42 | 87.92 ± 10.82 | 86.49 ± 1.06 | 88.27 ± 0.09 |
| distilbert-ERM | 89.30 ± 6.15 | 97.63 ± 0.49 | 86.92 ± 3.35 | 95.08 ± 0.99 | 97.13 ± 0.96 | 90.79 ± 5.93 | 89.17 ± 8.24 | 93.59 ± 0.87 | 92.17 ± 0.07 |
| distilbert-ERM-seed1 | 89.10 ± 7.12 | 97.48 ± 0.50 | 87.05 ± 3.34 | 94.86 ± 1.05 | 97.14 ± 1.01 | 90.75 ± 6.16 | 95.07 ± 6.47 | 93.54 ± 0.83 | 92.17 ± 0.08 |
| distilbert-ERM-seed2 | 89.20 ± 6.34 | 98.10 ± 0.42 | 86.67 ± 3.43 | 95.78 ± 0.94 | 96.28 ± 1.23 | 90.43 ± 6.83 | 92.84 ± 6.25 | 93.65 ± 0.82 | 92.11 ± 0.08 |
| distilbert-IRM | 86.90 ± 7.62 | 93.03 ± 0.77 | 82.92 ± 3.26 | 95.27 ± 0.66 | 94.56 ± 2.34 | 88.11 ± 7.22 | 88.10 ± 9.13 | 91.65 ± 0.97 | 88.13 ± 0.09 |
| distilbert-IRM-seed1 | 88.10 ± 7.35 | 94.03 ± 0.81 | 83.86 ± 3.31 | 95.92 ± 0.67 | 95.41 ± 1.76 | 89.27 ± 7.03 | 89.42 ± 9.13 | 92.26 ± 0.84 | 89.04 ± 0.10 |
| distilbert-IRM-seed2 | 86.80 ± 6.91 | 93.49 ± 0.80 | 83.39 ± 3.24 | 95.55 ± 0.69 | 95.34 ± 2.18 | 87.78 ± 7.13 | 89.19 ± 9.26 | 92.08 ± 0.91 | 88.70 ± 0.08 |

Table S8: **Mean absolute error in accuracy estimation per classifier on toxicity detection.** .

| model | Labeled | Majority-Vote | Pseudo-Labeling (LR) | Dawid-Skene | Bayesian-Calibration | SSME-KDE (Ours) | Ground Truth |
|---|---|---|---|---|---|---|---|
| distilbert-CORAL | 13.80 ± 5.81 | 8.27 ± 0.79 | 8.78 ± 3.60 | 10.78 ± 1.08 | 7.47 ± 4.14 | 8.50 ± 0.95 | 5.98 ± 0.09 |
| distilbert-ERM | 10.37 ± 5.86 | 1.38 ± 0.44 | 11.33 ± 3.29 | 4.37 ± 1.06 | 1.97 ± 1.01 | 4.96 ± 0.75 | 6.14 ± 0.09 |
| distilbert-ERM-seed1 | 9.91 ± 6.29 | 1.57 ± 0.38 | 11.29 ± 3.24 | 4.59 ± 1.13 | 1.93 ± 1.07 | 5.13 ± 0.78 | 6.21 ± 0.08 |
| distilbert-ERM-seed2 | 10.02 ± 6.02 | 0.57 ± 0.32 | 10.96 ± 3.33 | 3.38 ± 0.99 | 2.32 ± 1.26 | 3.99 ± 0.71 | 4.94 ± 0.09 |
| distilbert-IRM | 12.59 ± 6.89 | 6.47 ± 0.70 | 15.77 ± 3.08 | 3.69 ± 0.67 | 4.61 ± 2.62 | 6.85 ± 0.88 | 10.61 ± 0.08 |
| distilbert-IRM-seed1 | 11.93 ± 6.79 | 5.37 ± 0.73 | 15.03 ± 3.26 | 2.98 ± 0.63 | 3.88 ± 1.99 | 6.38 ± 0.93 | 9.78 ± 0.11 |
| distilbert-IRM-seed2 | 12.06 ± 6.12 | 5.77 ± 0.75 | 15.55 ± 3.14 | 3.33 ± 0.74 | 3.87 ± 2.35 | 6.71 ± 0.89 | 10.18 ± 0.09 |

Table S9: **Mean absolute error in ECE estimation per classifier on toxicity detection.**

SSME's improvement in performance estimation can be attributed to a significant reduction in the variance of performance estimates across different data splits.

### D.3 COMPARISON TO BASELINES DRAWN FROM WEAK SUPERVISION

Popular approaches to weak supervision including Snorkel (Ratner et al., 2017) and FlyingSquid (Fu et al., 2020) implement a latent variable model equivalent to Dawid-Skene. Both works build on Dawid-Skene to incorporate information about pairwise correlations between labeling functions; (Ratner et al., 2017) employs a technique to infer dependencies, while (Fu et al., 2020) assume these dependencies to be user-provided. When we applied a standard approach to dependency inference (Bach et al., 2017) in our setting, we observed that (1) all classifiers are inferred to be dependent on one another, and (2) the number of dependencies raised issues with convergence. It is thus not feasible to incorporate dependency inference, and the resulting latent variable model is equivalent to Dawid-Skene.

### D.4 COMPARISON TO ENSEMBLING

While we limit the scope of our experiments in the main text to semi-supervised methods that make use of *both* labeled and unlabeled data, another approach would be to produce an estimate of $Pr(y = k|s^{(i)})$ by averaging the classifier scores. This approach results in an unbiased metric estimator when the resulting ensemble is calibrated, as theoretical results by Ji et al. (2020) show. Such an approach has natural downsides: it is sensitive to the composition of the classifier set, does not improve with the introduction of labeled data, and relies on an assumption of ensemble calibration that is unlikely to hold in practice (Wu & Gales, 2021). Here, we provide experiments to illustrate this behavior.

Using the semisynthetic setting described in Section 6.4, we artificially increase the expected calibration error of each classifier using a generalized logistic function parameterized by $a$. Specifically, we transform classifier score $s$ to be $\frac{s^a}{s^a+(1-s)^a}$, effectively increasing overconfidence for higher $s$ and increasing underconfidence for lower $s$. As in the semisynthetic experiments, we generate 500 semisynthetic classifier sets, where each classifier in a set is trained on 100 examples distinct from the training data for other classifiers in the set (results are robust to this choice of training dataset size). Each set contains three classifiers.

| model | Labeled | Majority-Vote | Pseudo-Labeling (LR) | Dawid-Skene | Bayesian-Calibration | SSME-KDE (Ours) | Ground Truth |
|---|---|---|---|---|---|---|---|
| distilbert-CORAL | 85.19 ± 14.15 | 95.09 ± 1.01 | 72.20 ± 6.89 | 94.89 ± 1.18 | 84.22 ± 3.49 | 91.38 ± 1.60 | 86.23 ± 0.17 |
| distilbert-ERM | 91.80 ± 7.11 | 99.29 ± 0.25 | 74.90 ± 7.69 | 98.64 ± 0.33 | 98.52 ± 0.94 | 95.97 ± 0.77 | 93.77 ± 0.11 |
| distilbert-ERM-seed1 | 92.18 ± 7.46 | 99.22 ± 0.26 | 74.92 ± 7.68 | 98.58 ± 0.30 | 98.30 ± 0.95 | 95.96 ± 0.71 | 93.75 ± 0.10 |
| distilbert-ERM-seed2 | 93.32 ± 5.93 | 99.47 ± 0.23 | 75.03 ± 7.74 | 98.89 ± 0.32 | 98.07 ± 1.12 | 96.16 ± 0.68 | 94.08 ± 0.10 |
| distilbert-IRM | 91.46 ± 8.74 | 98.41 ± 0.48 | 74.69 ± 7.49 | 98.28 ± 0.60 | 98.05 ± 0.96 | 95.38 ± 1.13 | 91.57 ± 0.13 |
| distilbert-IRM-seed1 | 90.75 ± 10.48 | 98.20 ± 0.81 | 74.58 ± 7.58 | 97.97 ± 0.81 | 98.07 ± 1.02 | 95.18 ± 1.12 | 91.00 ± 0.16 |
| distilbert-IRM-seed2 | 91.89 ± 8.13 | 98.40 ± 0.44 | 74.68 ± 7.60 | 98.39 ± 0.52 | 98.33 ± 0.94 | 95.59 ± 0.91 | 91.86 ± 0.11 |

Table S10: **Mean absolute error in AUC estimation per classifier on toxicity detection.**

| model | Labeled | Majority-Vote | Pseudo-Labeling (LR) | Dawid-Skene | Bayesian-Calibration | SSME-KDE (Ours) | Ground Truth |
|---|---|---|---|---|---|---|---|
| distilbert-CORAL | 60.78 ± 24.13 | 64.21 ± 5.46 | 31.91 ± 10.19 | 76.38 ± 5.00 | 50.86 ± 10.73 | 60.27 ± 5.50 | 40.00 ± 0.36 |
| distilbert-ERM | 77.25 ± 19.52 | 96.37 ± 1.09 | 42.28 ± 13.73 | 94.59 ± 1.22 | 92.51 ± 4.32 | 79.63 ± 3.68 | 72.19 ± 0.37 |
| distilbert-ERM-seed1 | 79.38 ± 18.74 | 96.06 ± 1.18 | 42.27 ± 13.65 | 94.37 ± 1.15 | 91.54 ± 4.94 | 79.78 ± 3.46 | 72.30 ± 0.35 |
| distilbert-ERM-seed2 | 79.11 ± 18.05 | 97.10 ± 1.01 | 42.47 ± 13.79 | 95.49 ± 1.19 | 90.15 ± 4.94 | 80.06 ± 3.34 | 73.33 ± 0.35 |
| distilbert-IRM | 77.74 ± 19.47 | 90.10 ± 2.39 | 40.79 ± 13.20 | 92.89 ± 1.47 | 88.23 ± 7.39 | 76.90 ± 4.22 | 65.86 ± 0.40 |
| distilbert-IRM-seed1 | 79.21 ± 19.99 | 91.23 ± 2.66 | 41.23 ± 13.47 | 93.64 ± 1.62 | 89.58 ± 5.67 | 77.84 ± 3.91 | 66.50 ± 0.44 |
| distilbert-IRM-seed2 | 77.63 ± 20.39 | 89.37 ± 2.40 | 40.65 ± 13.36 | 92.67 ± 1.77 | 90.34 ± 6.36 | 77.16 ± 3.67 | 65.46 ± 0.40 |

Table S11: **Mean absolute error in AUPRC estimation per classifier on toxicity detection.**

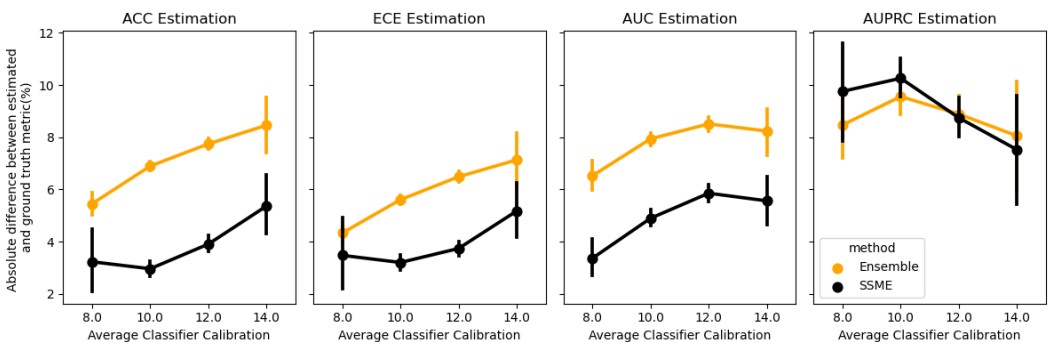

Figure S3: **A comparison of SSME to ensembling on a miscalibrated classifier set.** SSME consistently produces more accurate performance estimates compared to ensembling the classifiers across differently calibrated classifier sets (x-axis).

Figure S3 reports our results. As the average calibration among classifiers in a set varies, SSME consistently improves over the use of an ensemble. This aligns with our intuition, and indicates the value of using labeled data in conjunction with unlabeled data. Interestingly, miscalibration has little effect on the ensemble when estimating AUPRC; here, SSME and ensembling perform similarly.

### D.5 DISCUSSION OF CLASSIFIER CORRELATION

SSME, in contrast to prior work, makes no asumption about the correlation between classifiers because any assumption is unlikely to hold in practice. The average correlation between classifiers in our sets for each binary task is 0.53, 0.85, 0.93, 0.81, 0.77 (for ED revisit, critical outcome, hospitalization, toxicity, and SARS-COV inhibition prediction respectively). This range of values reflects natural correlation between classifiers in practice, since each of our models is either an off-the-shelf classifier or trained using publicly available code.

## E METHOD DETAILS

### E.1 METRIC ESTIMATION

Given a vector $p \in \Delta^{K-1}$ over $K$ classes, let $\mathbf{s} = \mathrm{ALR}(\mathbf{p}) = \left[\log \frac{\mathbf{p}_1}{\mathbf{p}_K}, \log \frac{\mathbf{p}_2}{\mathbf{p}_K}, \cdots, \log \frac{\mathbf{p}_{K-1}}{\mathbf{p}_K}\right] \in \mathbb{R}^{K-1}$. To invert, $\mathbf{p}_i = \frac{e^{\mathbf{s}_i}}{1+\sum_{k=1}^{K-1} e^{\mathbf{s}_k}}$ for $i < K$ and $\mathbf{p}_K = \frac{1}{1+\sum_{k=1}^{K-1} e^{\mathbf{s}_k}}$. The ALR transform maps unit-sum data into real space, where it is easier to fit mixture models. The inverse allows us to map samples from the mixture model in real space back to the simplex $\Delta^{K-1}$. For details, see Pawlowsky-Glahn & Buccianti (2011).

### E.2 METRIC ESTIMATION

SSME is able to estimate any metric that is a function of the classifier probabilities $p$ and label $y$. We approximate the joint distribution $P(y, \mathbf{p})$ with a mixture model model $P_\theta(y, \mathbf{s})$, where $\mathbf{s}$ refers

to the ALR-transformed classifier probabilities (i.e. "classifier scores")[2]. We refer to $P(y, \mathbf{p})$ for ease of notation in this section; it is equivalent, through invertible mapping, to $P(y, \mathbf{s})$.

We denote our approximation for $P(\mathbf{p}, y)$ as $P_\theta(\mathbf{p}, y)$. We provide a few concrete examples of how one can use SSME to measure performance metrics, given $P_\theta(\mathbf{p}, y)$ and a set of unlabeled probabilistic predictions $\{\mathbf{p}^{(i)}\}_{i=1}^{n_u}$ and labeled probabilistic predictions $\{\mathbf{p}^i, y^{(i)}\}_{i=1}^{n_\ell}$. Notationally, $\mathbf{p}_j^i$ refers to the $j$th model's probabilistic prediction of the $i$th unlabeled example.

**Accuracy** measures the alignment between a model's (discrete) predictions and the true label $y$. To discretize predictions, practitioners typically take the argmax of $\mathbf{p}^{(i)}$. Using the binary case an illustrative example, the accuracy of the $j$th model can be written as:

$$\text{Accuracy}_j = \mathbb{E}_\mathbf{p}\left[\mathbf{1}\left[y = \mathbf{1}(\mathbf{p} > t)\right]\right]$$

where $\mathbf{1}$ is an indicator function and $t$ is a chosen threshold, typically 0.5. In our setting, we approximate this as:

$$\text{Accuracy}_j \approx \frac{1}{n_u + n_\ell} \sum_{i=1}^{n_u+n_\ell} \mathbf{1}\left[y^{(i)} = \mathbf{1}(\mathbf{p}^{(i)} > t)\right]$$

For labeled examples, we use the true label $y^{(i)}$. For unlabeled examples, we draw $y^{(i)} \sim P_\theta(y|\mathbf{p}^{(i)})$. We then compute accuracy using these labels $y^{(i)}$ and predictions $\mathbf{p}^{(i)}$. To ensure our estimation procedure is robust to sampling noise, we average our estimated accuracy over 500 separate sampled labels for each example in the unlabeled dataset.

Alternatively, we could directly use $P_\theta(y|\mathbf{p})$ to estimate accuracy. That is, for each point $\mathbf{p}^{(i)}$ we directly compute an expectation for the label, and sum this over the entire dataset.

Using the binary case as an example

$$\text{Accuracy}_j \approx \frac{1}{n_u + n_\ell} \sum_{i=1}^{n_u+n_\ell} \mathbb{E}\left[\mathbf{1}\left[y^{(i)} = \mathbf{1}(\mathbf{p}_j^{(i)} > t)\right] | \mathbf{p}^{(i)}\right]$$

In other words, we compute the expectation that the true label agrees with the predicted label for each point . This expectation is $\mathbf{p}^{(i)}$. This expectation is computed over $P_\theta(y|\mathbf{p})$ One can interpret $P_\theta(y|\mathbf{p})$ as a "recalibration" step: given a set of classifier guesses $\mathbf{p}$, what is the true distribution of $y$?

In our experiments, we use the first of these two approaches, i.e. we sample the true label from the estimated distribution.

**Expected Calibration Error (ECE)** measures the alignment between a model's predicted probabilities $\mathbf{p}_j$ and the ground truth labels $y$. In particular, ECE compares the model's reported confidence to the true class likelihoods, averaged over the dataset. We write out our ECE estimation procedure for the binary case, and it extends readily to definitions of calibration in multiclass settings (Gupta & Ramdas, 2022). Binary ECE can be written as:

$$\text{ECE}_j = \mathbb{E}_{\mathbf{p}_j}\left[\left|P(\hat{Y} = 1|\hat{p} = \mathbf{p}_j) - \mathbf{p}_j\right|\right]$$

Then, to approximate the ECE with the datasets $\{\mathbf{p}^i\}_{i=1}^{n_u}$ and $\{\mathbf{p}^i, y^{(i)}\}_{i=1}^{n_\ell}$, one can sample $y^{(i)} \sim P_\theta(y|\mathbf{p}^{(i)})$ for each unlabeled sample $i$ and then use the standard histogram binning procedure (Guo et al., 2017) using both the true labels for the labeled dataset and the sampled labels for the unlabeled dataset. In this approach, we treat the sampled labels $y^{(i)}$ as true labels for unlabeled examples. To ensure our procedure is robust against sampling noise, we draw samples of $y^{(i)}$ repeatedly for a fixed number of draws (500). We then compute ECE separately for each of these 500 draws and average ECE across all draws.

---

[2]Recall that ALR is a bijection, so we use the inverse mapping $\text{ALR}^{-1} : \mathbb{R}^{K-1} \to \Delta^{K-1}$ to transform our mixture distribution in real space back to probability space.

Alternatively, one could also *directly* use $P_\theta(y|\mathbf{p})$ to estimate ECE. In particular, we can write:

$$\text{ECE}_j \approx \frac{1}{n_u + n_\ell} \sum_{i=1}^{n_u + n_\ell} \left| P_\theta\left(y = 1|\mathbf{p}_j^{(i)}\right) - \mathbf{p}_j^{(i)} \right|$$

In this approach, we don't sample the labels $y$ for unlabeled examples but instead directly use $P_\theta(y|\mathbf{p})$, which provides us (an estimate of) the true distribution of $y$. Instead, we directly use our estimate for the conditional label distribution $P_\theta\left(y = 1|\mathbf{p}_j^{(i)}\right)$. In our experiments, we use the first approach described, i.e. sampling $y^{(i)}$ for unlabeled examples and then using the standard binning and averaging procedure.

**AUROC and AUPRC** can be estimated with a similar procedure as above. In particular, we sample a label $y^{(i)} \sim P_\theta\left(y = 1|\mathbf{p}^{(i)}\right)$ from the conditional label distribution and compare these sampled labels to the classifier probabilities.

### E.3   THEORETICAL INSIGHTS INTO SSME

Since SSME is a semi-supervised learning method, we can gain theoretical insights into its performance by drawing from results in semi-supervised learning theory. We summarize the data and modeling assumptions, backed by prior theoretical work, under which SSME is likely to succeed (or fail). For a full survey of the theory of semi-supervised learning, see Mey & Loog (2022).

**Data assumptions**: SSME will perform better when two common semi-supervised learning assumptions are met in the data:

1. **Smoothness**: examples $i, j$ with similar classifier scores $\mathbf{s}^{(i)}, \mathbf{s}^{(j)}$ (for a suitable notion of similar) are likely to share the same true label $y^{(i)}, y^{(j)}$.

2. **Clusters and low-density separation**: classifier scores $\mathbf{s}$ cluster according to their true classes $y$, and high-density cluster centers are separated by low-density regions. These low-density regions can help identify decision boundaries, even in the absence of large, labeled datasets.

In our setting, assumption (2) can be particularly helpful. If, for instance, in a $K = 3$ class classification problem, we observe that most classifier scores $\mathbf{s}$ cluster into one of three corners of the 3-simplex (with low density in between the corners), it is likely that each of these clusters corresponds to a different class.

Prior work has shown that when the above two assumptions are met, semi-supervised learning works well (Singh et al., 2008). Singh et al. (2008) formalizes these assumptions and shows that if mixture component densities and boundaries are discernable from $m$ unlabeled examples but not from $n < m$ labeled examples, then semi-supervised learning can improve performance relative to labeled data alone. They characterize this relationship and show that the error between the estimated density and true density is reduced by a function of $n, m$, and the data dimension $d$ when using semi-supervised learning compared to labeled data alone. Our empirical results also substantiate this result, as we show that SSME outperforms metric estimation using labeled data alone.

**Modeling assumptions**: SSME is more likely to succeed when the mixture densities are well-specified. In other words, if the true underlying density is from the same class of distributions as the parameterized mixture densities (e.g., Gaussian density), then adding unlabeled data enables more accurate estimates of the components relative to using labeled data alone (Cozman et al.). If, on the other hand, the mixture distribution is not well-specified, unlabeled data may hurt. Cozman et al. formalize this analysis, including under what conditions semi-supervised mixtures are robust to mis-specification. Our default parameterization, a KDE, is flexible and can accommodate a wide variety of distributions.

Even when the data assumptions are met, the model complexity and performance will likely follow the classic bias-variance tradeoff. Simpler models with fewer parameters (e.g., Dawid-Skene) are likely to yield biased estimates of the true density (and thus calculated metrics like accuracy) while more complex models will yield less bias at the cost of greater variance. Our parameterization,

KDEs, have few parameters (other than the bandwidth), unlike approaches which explicitly model the joint density from a particular distribution (e.g., Gaussian mixture model).

### E.4 EM ALGORITHM

We use the EM algorithm to fit SSME, which iterates between the $E$-step and $M$-step updates, described below.

**E-step**: For the $t$th update, we compute:

$$P_\theta^{t+1}(y^{(i)} = k|\mathbf{s}^{(i)}) = \frac{p^t(y = k)P_\theta^t(\mathbf{s}^{(i)}|y^{(i)} = k)}{\sum_{\ell=1}^K p^t(y = \ell)P_\theta^t(\mathbf{s}^{(i)}|y^{(i)} = \ell)}$$

This is combined with the prior $p^t(y = k)$ to produce posteriors $P_\theta^{t+1}(y^{(i)=k|\mathbf{s}^{(i)}})$. We then fix the labels for labeled example, setting $P_\theta(P_\theta(y^{(i)=k^*|\mathbf{s}^{(i)}}) = 1$ for the correct class $k*$ and to 0 for all other classes.

**M-step**: During the M-step, we update the prior $p^t(y = k)$. To do so, we calculate:

$$p^{t+1}(y = k) = \frac{1}{n_u + n_\ell}\left(\sum_{i=1}^{n_\ell} I(y^{(i)} = k) + \sum_{i=1}^{n_u} P_\theta^{t+1}(y^{(i)} = k|\mathbf{s}^{(i)})\right)$$

We alternate between the above two updates for 1000 epochs.

