# OpenReview forum: "Evaluating multiple models using labeled and unlabeled data"
_ICLR.cc/2025/Conference — Submitted to ICLR 2025_

### Official Review · Reviewer_w43e · 2024-10-29

**Soundness:** 2
**Presentation:** 1
**Contribution:** 1
**Rating:** 6
**Confidence:** 3

**Summary:**

The paper studies a practical setting which is to evaluate the performance of $M$ classifiers pre-trained targetting the same tasks on two subsets, one with labels and the other without labels. The solution proposed in the paper is to employ semi-supervised learning following a generative modelling through a mixture model to estimate the joint probability of the ground truth labels of unlabelled data and the predictions made by those $M$ classifiers. The ground truth is them inferred from such a modelling to evaluate the performance. The empirical experiment presented in the paper covers a number of datasets.

**Strengths:**

The paper introduces a setting often encountered in practice: evaluating performance of $M$ classifiers pre-trained for a task on a new dataset. In addition, that dataset has a small portion labelled, while the remaining is unlabelled. The paper is well-written to explain the idea at high-level.

**Weaknesses:**

Despite the easy-to-understand being of the paper at high level, the paper poorly explains how the main idea can be done in technical details (including the text in section 4 and the one in Appendix C). The explanation is too vague for me to understand how the paper is doing to achieve its goal.

According to the problem setting in section 3, the problem is equivalent to infer the ground truth labels on unlabelled data using the labelled data and some additional pre-trained classifiers. If one knows the distribution of the ground truth for each sample, it is easier to evaluate the performance of those pre-trained classifiers. Stating this equivalence would significantly improve the clarity of the paper.

The main idea of the paper is to follow the generative approach in semi-supervised learning to find the joint probability between label $y$ and another variable (in the paper, it is the concatenated predictions of all the classifiers, denoted as $\mathbf{s}$). The equation at line 167 can be rewritten in a more understandable form as follows:

$$
\max_{\theta} \ln \Pr(S, Y, S^{\prime}; \theta) =  \max \ln \prod_{i = 1}^{n_{l}} \Pr(\mathbf{s}\_{i}, y\_{i}; \theta) \prod_{j = 1}^{n\_{u}} \Pr(\mathbf{s}\_{j}; \theta) = \max \sum_{i = 1}^{n\_{l}} \ln \Pr_{\theta}(\mathbf{s}\_{i}, | y\_{i}) \Pr(y\_{i}) + \sum_{j = 1}^{n\_{u}} \ln \sum_{k = 1}^{K} \Pr_{\theta}(\mathbf{s}\_{j} | y\_{j} = k) \Pr(y\_{j} = k).
$$

Here, it is not explained why the vector concatenated predictions of all the classifiers, $\mathbf{s}$, is used as one of the variables, instead of the raw input data $\mathbf{x}$ as in conventional semi-supervised learning. This may lead to a bad estimation if most of the classifiers are bad. My guess is that the setting does not allow to access to the raw data $\mathbf{x}$. However, in section 3, the raw input data $\mathbf{x}$ is accessible.

Another concern is how to know the label distribution $\Pr(y\_{j})$ of the unlabelled data (the very last term in the equation above). In the text, the authors only explain how to calculate $\Pr(\mathbf{s} | y)$ at line 181, without specifying how to calculate $\Pr(y\_{j})$ of the unlabelled data term. This distribution must be known to estimate $\Pr(y | \mathbf{s}$ from the learnt $\Pr(\mathbf{s}, y)$, which is claimed by the paper, but is never explained how to perform such inference.

**Questions:**

My concern is the main method proposed to infer the ground truth labels of unlabelled data. Could the authors clarify how such ground truth labels are inferred in more details? The current explanation is too vague to understand.

In addition, could the authors clarify why using $\mathbf{s}$ instead of $\mathbf{x}$ in the formation?

---

> ### Author Response · Authors · 2024-11-20
> **Thanks for your feedback!**
>
> Thank you for the comments. We are glad you found the paper’s main ideas easy to understand and raised no significant empirical/experimental concerns. We apologize that some of the methodological details were unclear to you and seek to rectify this. To respond to your concerns about methodological clarity, we first summarize and clarify the key ideas of our method; we then provide point-by-point responses to specific questions and highlight changes we’ve made to the manuscript to address them.
>
> **Summary of SSME**
> Our method performs simultaneous evaluations of multiple classifiers using both labeled and unlabeled data. As you note, if every datapoint were labeled, we could compare classifier predictions to these labels to estimate common metrics such as accuracy and calibration. In practice, however, labels are often expensive or impossible to obtain for unlabeled examples. We infer a distribution over these missing ground truth labels and use these inferred labels on unlabeled examples along with true labels on labeled examples to conduct our evaluations. More specifically, SSME infers labels by learning the joint distribution p(y, s) over classifier scores (i.e., classifier probabilities), s, and true labels, y. We fit a mixture model to estimate this joint distribution p(y, s), with a separate mixture component per class. Intuitively, for a K-class classification problem, classifier predictions should “cluster” into different corners of the K-simplex, provided the classifiers are better than random. We model the joint distribution of scores s with y instead of the raw inputs x with y.
>
> **Q1: Restating equation at Line 167 more clearly**
>
> Thank you for this suggestion! We have edited the equation according to your suggested rewrite (Line 190 in the revised draft) and believe it to be clearer now. Let us know if you have suggestions or requests for further clarification!
>
> **Q2: Explaining relation to supervised evaluation**
>
> Good idea - we have included a statement to this effect at Lines (148-150) to clarify that, if we had access to labels for each example, classifier evaluation would be straightforward. Since we do not, we must instead infer the labels for the unlabeled examples, which SSME is designed to do.
>
> **Q3: How are ground truth labels inferred?**
>
> We first estimate our joint distribution p(y,s) using the EM algorithm. During EM, we iterate between the E-step, during which we compute soft assignments of which mixture component each example belongs to, and the M-step, during which we update the parameters of each mixture component (and thus, our estimates of p(y) and p(s|y)) based on the soft component assignments.
>
> Once EM has terminated, we infer ground truth labels for unlabeled examples by drawing labels several times according to p(y|s). These inferred labels are then used to estimate metrics such as accuracy, calibration, etc. For labeled examples, we use the true label during metric estimation. We have now clarified our ground truth label inference procedure in a new “Evaluation” subsection of Section 3 (Lines 240-246).
>
> **Q4: How is P(y) in the unlabeled data inferred?**
>
> Good question. We follow standard practice in expectation-maximization and estimate p(y) during the M-step using the soft estimated cluster assignments. Specifically, we estimate p(y=y_k) by averaging the mean probability that an example is assigned to class y_k across all examples. We have added a full description of our EM algorithm and fitting procedure in Appendix E.4.
>
> **Q5: Why do we use s instead of x?**
>
> It is a standard decision in many semi-supervised evaluation settings to model the joint distribution p(y, s) as opposed to p(y, x) (see e.g., Ji et al. 2020, Boyeau et al. 2024, Welinder et al. 2013, Chouldechova et al. 2022). There are a few reasons for this choice:
> Fitting p(s|y) is easier than fitting p(x|y): x can be very high-dimensional (e.g., if it is an image) whereas s is typically lower-dimensional.
> Modeling p(s, y) is sufficient to estimate any metric that is a function of a model's predictions and ground truth labels.
> We also follow a rich literature in using s instead of x (Bayesian calibration, AutoEval, Dawid-Skene) and improve on it by expanding s to include multiple classifiers.
>
> **We have added a paragraph to Section 4** (Lines 164-170) to explain in more detail why we choose to model s instead of x.
>
> Please let us know if our response addresses your primary concerns, or if there is further clarification of our method that would be useful!
>
> [Ji et al. 2020] https://arxiv.org/abs/2010.09851
>
> [Boyeau et al. 2024] https://arxiv.org/abs/2403.07008
>
> [Welinder et al. 2013] https://ieeexplore.ieee.org/abstract/document/6619263
>
> [Chouldechova et al. 2022] https://www.amazon.science/publications/unsupervised-and-semi-supervised-bias-benchmarking-in-face-recognition

---

> > ### Comment · Reviewer_w43e · 2024-11-21
> > **Contributions and importance**
> >
> > It is clear that the paper is to propose to use the EM algorithm to perform a semi-supervised learning following a generative approach. However, there are a few concerns that should be clarified further:
> >  - The main method is the EM algorithm. However, it was not specified explicitly in the original submission. Furthermore, it is now presented in the Appendices, which often means that it is not important, although that is the main method presented in the paper. Doing that would not make the paper ready for publication.
> >  - The usage of the EM algorithm in semi-supervised learning has been studied in the literature. In addition, the one presented in the paper does not provide any new insights nor modification, except it is applied to the current setting. Thus, it limits the contribution of the paper. Could the authors elaborate the contributions of the paper besides all the empirical evaluation?
> >  - The authors have not addressed my previous concern that what happens when most of the classifiers are bad. In that case, the ground truth inferred may not be the true ground truth. Hence, it is meaningless to proceed further.
> >  - Could the authors provide further explanations on the setting? In the current setting, the ground truth of unlabelled data is inferred from the labelled data and **the prediction of classifiers** using EM under the formulation of semi-supervised learning. In other words, it is analogous to minimise the entropy, meaning that the ground truth inferred tends to agree with the prediction of classifiers. This contradicts to the standard evaluation because the ground truth should be independent (or marginalised out over all (infinite number of) classifiers). Hence, to my current understanding, the study seems to be not useful for evaluation.

---

> > > ### Author Response · Authors · 2024-11-21
> > > **Additional clarification**
> > >
> > > Thanks for continuing to engage with our work; we appreciate the quick response.
> > >
> > > **The main method is the EM algorithm. However, it was not specified explicitly in the original submission. Furthermore, it is now presented in the Appendices, which often means that it is not important, although that is the main method presented in the paper. Doing that would not make the paper ready for publication.**
> > >
> > > Thanks for this comment! To clarify, the original submission did specify and describe the use of the EM algorithm in the main text (Lines 226-230). The equations in the appendix make these statements more formally. However, we are happy to move them into the main text.
> > >
> > > Importantly, the main novel contribution of our work is a new method for semi-supervised multi-classifier evaluation. Like many novel methods, ours relies on EM as an optimization procedure; for other examples of novel methods relying on EM, see e.g., Jang et al. 2016, Cui et al. 2019, and Jin et al. 2022.
> > >
> > > Jang et al., 2016 - https://arxiv.org/abs/1605.08174
> > >
> > > Cui et al., 2019 - https://arxiv.org/pdf/1903.02419
> > >
> > > Jin et al., 2022 - https://arxiv.org/abs/2211.11427
> > >
> > >
> > > **The usage of the EM algorithm in semi-supervised learning has been studied in the literature. In addition, the one presented in the paper does not provide any new insights nor modification, except it is applied to the current setting. Thus, it limits the contribution of the paper. Could the authors elaborate the contributions of the paper besides all the empirical evaluation?**
> > >
> > > Thank you! To clarify, our contribution is not to advance methods in semi-supervised learning or to present EM as a novel solution. Instead, our contribution is to propose a new method to evaluate multiple classifiers in the presence of limited labeled data. SSME is the first evaluation method to simultaneously learn from: (i) multiple machine learning classifiers, (ii) probabilistic predictions over all classes, and (iii) unlabeled data – and we show empirically that this improves results.
> > >
> > > **The authors have not addressed my previous concern that what happens when most of the classifiers are bad. In that case, the ground truth inferred may not be the true ground truth. Hence, it is meaningless to proceed further.**
> > >
> > > Thank you! We find that SSME performs well empirically even when classifiers are not strong. For example, the AUCs of the classifiers trained to predict ED revisit range from 0.55 to 0.60. Even here, SSME is able to produce performance estimates that (1) are more accurate than those of the baselines and (2) are reasonably close to ground truth performance. SSME does perform better with stronger classifiers; however, this is not unique to SSME, but rather true of all semi-supervised evaluation methods (for example, Welinder et al. 2013, Ji et al., 2020, Boyeau et al., 2024).
> > >
> > > **Could the authors provide further explanations on the setting? In the current setting, the ground truth of unlabelled data is inferred from the labelled data and the prediction of classifiers using EM under the formulation of semi-supervised learning. In other words, it is analogous to minimise the entropy, meaning that the ground truth inferred tends to agree with the prediction of classifiers. This contradicts to the standard evaluation because the ground truth should be independent (or marginalised out over all (infinite number of) classifiers). Hence, to my current understanding, the study seems to be not useful for evaluation.**
> > >
> > > The reviewer is exactly correct that “In the current setting, the ground truth of unlabelled data is inferred from the labelled data and the prediction of classifiers using EM under the formulation of semi-supervised learning.”
> > > This is in fact a standard procedure in semi-supervised evaluation  [Ji et al., 2020; Boyeau et al., 2024], where prior work has established that classifier predictions — even if noisy — can help perform evaluations with fewer ground truth labels.
> > >
> > > To address the reviewer’s concern that our approach “seems to be not useful for evaluation”: our results across 5 datasets demonstrate that using unlabeled data and classifier predictions is useful for evaluation, as together these improve estimates of accuracy compared to using the limited labeled data alone.
> > >
> > > [Welinder et al., 2013] https://ieeexplore.ieee.org/abstract/document/6619263
> > >
> > > [Ji et al., 2020] https://arxiv.org/pdf/2010.09851
> > >
> > > [Boyeau et al., 2024] https://arxiv.org/abs/2403.07008

---

> > > > ### Comment · Reviewer_w43e · 2024-11-24
> > > > **Concern on overfitting**
> > > >
> > > > Thank you, the authors, for clarification. It is indeed what I understand about the paper. My main concern is that there is an overfitting because ground truth labels of unlabelled samples are inferred from labelled data and the prediction of classifiers on unlabelled data. In other words, the inferred ground truth would most likely agree with the prediction of the classifiers. The three splits mentioned in section 5.3 is indeed an empirical Bayes way to avoid such overfitting. I missed this part during my initial review. I, therefore, update my rating for the paper.

---

> > > > > ### Author Response · Authors · 2024-11-24
> > > > > **Thank you!**
> > > > >
> > > > > Thank you for closely reviewing our work, and for updating your score! We appreciate it.

---

### Official Review · Reviewer_SyCL · 2024-11-02

**Soundness:** 2
**Presentation:** 2
**Contribution:** 2
**Rating:** 5
**Confidence:** 4

**Summary:**

This paper introduces **Semi-Supervised Model Evaluation (SSME)**, a method that leverages both labeled and unlabeled data to evaluate multiple machine learning classifiers. SSME is built on the premise that unlabeled data is often more abundant than labeled data,  addressing scenarios where multiple classifiers are available but labeled data is scarce. Using a semi-supervised mixture model, SSME estimates the joint distribution of ground truth labels and classifier scores, which allows it to assess metrics such as accuracy and expected calibration error  without requiring extensive labeled datasets. Through experiments in healthcare, content moderation, molecular property prediction, and image annotation, SSME is shown to reduce estimation error in the metrics compared to baselines.

**Strengths:**

1. The paper presents a novel approach to evaluate classifiers by leveraging unlabeled data alongside limited labeled data, addressing a common bottleneck in evaluating models. The method is simple and the paper is well written overall.

2. SSME’s performance is evaluated across multiple domains and tasks, such as healthcare and content moderation.  The results indicate improvements over baselines in estimating standard metrics, providing evidence of the method's applicability.

3.  The study also shows methods ability to evaluate subgroup-specific metrics, which is beneficial for fairness assessments. This application is particularly relevant for sensitive domains like healthcare, where performance disparities among demographic groups can be critical.

**Weaknesses:**

1.  There is no theoretical analysis provided,  limiting our understanding of when and why the method may succeed or struggle in different data distributions or model configurations e.g. high or low accuracy models.

2.  The main empirical results in Figures 2, 3 and Table 1, report absolute error in estimates. I'd like to see the actual estimates, I believe if the models are highly accurate then it would be easier to infer the groundtruth label and hence the estimates are expected to be better but for models with low accuracy, or models with less correlation the observations could be different.

3. There are several baselines that have not been considered for evaluation. Some of them are based on weak supervision and active testing. You can use the same amount labeling budget as $n_l$ to run active testing and use $n_l$ points to estimate source quality or write labeling functions in weak supervision pipelines. See some of the references below,

https://arxiv.org/pdf/1711.10160

https://arxiv.org/abs/2107.00643

https://arxiv.org/abs/2002.11955

https://arxiv.org/pdf/2211.13375

https://arxiv.org/abs/2103.05331

**Questions:**

See the weakness above. I have a few more questions,

1. The estimated joint model is specific to the number of classifiers. So if one comes up with a new classifier how is that going to be evaluated? This modeling choice does not seem flexible to me.

2. The assumptions on the classification models are not clear. In particular, how accurate should they be and what's the correlation between them?

3. How will this approach work if the user gives classifiers one at a time, i.e. you don't see all the classifiers ahead of the time but instead they arrive sequentially. For instance, during model training we iteratively improve a single classifier and even during the developement cycle we maintain and update a single model.

---

> ### Author Response · Authors · 2024-11-20
> **Thanks for your feedback!**
>
> **W1: Theoretical grounding of SSME’s performance**
>
> Thank you for this question! We agree that the work would benefit from a deeper theoretical grounding of when SSME should succeed or fail. We provide this below, and have also added this discussion to the manuscript. Because SSME is a semi-supervised density estimation method, we draw on theoretical results in semi-supervised learning (SSL) theory. We summarize the data and modeling assumptions, backed by prior theoretical work, under which SSME is likely to succeed (or fail). We have added a new subsection E.3 to the Appendix with the discussion below.
>
> Data assumptions: SSME will perform better when the data it is fit on satisfies two common SSL assumptions:
> (1) the smoothness assumption: examples with similar classifier scores are likely to share the same label.
> (2) the cluster and low-density separation assumption: classifier scores cluster according to their (true) classes, and high-density cluster centers are separated by low-density regions (i.e., classifier score regions with few examples). These low-density regions can help identify the boundaries between classes.
>
> Prior work has shown that when these assumptions are met, SSL works well (Singh et al. 2008). Singh et al. formalize these assumptions and show that if mixture component densities and boundaries are discernable from m unlabeled examples but not from n < m labeled examples, then SSL can improve performance relative to labeled data alone. They characterize this relationship and show that the error between the estimated density and the true density is reduced by a function of n, m, and the data dimension d when using SSL compared to using labeled data alone. Our empirical results also substantiate this result; we find that SSME performs better when classifiers have higher accuracy, as scores from classifiers with high accuracy cluster better.
>
> Modeling assumptions: SSME is more likely to succeed when its model of the mixture densities is well-specified. In other words, if the true underlying density lies within the family of distributions that SSME can estimate, then adding unlabeled data enables more accurate estimates of the components relative to using labeled data alone (Cozman et al. 2003). If, on the other hand, the mixture distribution is not well-specified, unlabeled data may hurt. Cozman et al. formalize this analysis, including under what conditions semi-supervised mixtures are robust to mis-specification. Our default parameterization, a KDE, is flexible and can accommodate a wide variety of distributions, making it more likely it can capture the true mixture densities.
>
> In general, the model complexity and performance will likely follow the classic bias-variance tradeoff. Simpler models with fewer parameters (e.g., a Gaussian mixture model) are likely to yield biased estimates of the true density (and thus biased estimates of metrics like accuracy) while more complex models will yield less bias at the cost of greater variance. While we choose a particular parameterization (the KDE, which works well in our empirical settings), we note that our framework is flexible – e.g., as we show, it can also be implemented using a normalizing flow – and can accommodate other parameterizations.  SSME allows the user to choose a point on the bias-variance curve by selecting a particular parameterization.
>
> [Cozman et al. 2003] https://cdn.aaai.org/ICML/2003/ICML03-016.pdf
> [Singh et al. 2008] https://www.cs.cmu.edu/~aarti/pubs/NIPS08_ASingh.pdf
>
> **W2: Are results different for models that are less accurate, or less correlated with the other classifiers?**
>
> That’s a good question – our experiments suggest that SSME is able to estimate performance accurately for even the less accurate and less correlated classifiers in the set. SSME’s absolute estimates of AUPRC for each toxicity detection (Table S11 in the updated manuscript), for example, are more accurate and have lower variance compared to estimates from the baselines. SSME is able to do so because it uses information from other, more accurate classifiers to infer the labels for the unlabeled data. Baselines which do not use information from multiple models (e.g. Bayesian-Calibration) yield estimates further from ground truth.
>
> To make this point clearer, we have added Tables S8 - S11 to the manuscript, which describe absolute performance estimates in the context of toxicity detection (the task for which classifier quality varies the most), along with discussion of these results in Section D.3. We additionally report the ground truth performance of each classifier in Table S1 (binary classifiers) and Table S2 (multiclass classifiers). The tables illustrate the range of accuracies, ECEs, AUCs, and AUPRCs within each classifier set; for example, ground truth ECEs range from 0.06 to 0.11 on toxicity detection, while ground truth AUPRCs range from 0.40 to 0.72.

---

> > ### Author Response · Authors · 2024-11-20
> > **(continuation of response)**
> >
> > **W3: Comparison to baselines drawn from weak supervision literature**
> >
> > Thanks for the pointers to work on weak supervision and active testing! We agree there are important connections between these areas and our work. We respond in detail to each of the linked papers below:
> >
> > Suggested baseline 1 (https://arxiv.org/abs/2103.05331): Thanks for suggesting a comparison to active testing; we agree it is an important point of reference and have added it as a baseline to all experiments (Figure 2, Table 1). We note that active testing has an advantage over SSME: it actively selects the points to be labeled, while SSME assumes the labeled set is fixed. In spite of this, we find that SSME outperforms active testing across all tasks because it leverages predictions on the unlabeled data to measure performance, while active testing does not. Finally, we note that one could in principle use SSME to facilitate active testing, because SSME naturally provides estimates of label uncertainty for each datapoint; as such, SSME is a natural complement to an active testing approach, representing an interesting direction for future work.
> >
> > Suggested baselines 2 (Snorkel, https://arxiv.org/pdf/1711.10160) and 3 (FlyingSquid, https://arxiv.org/abs/2002.11955): Dawid-Skene (one of our baselines) is equivalent to both these baselines and thus, our experiments do capture a comparison to standard approaches in weak supervision. We have edited the draft to make the equivalence between these methods, and our work’s connections to weak supervision, more clear by adding Section D.3. We found that procedures that relax assumptions of conditional independence (for example, identifying sets of conditionally independent sources instead of assuming conditional independence between all sources) failed to converge in our setting, since classifier predictions are rarely independent conditional on class.
> >
> > Suggested baseline 4 (https://arxiv.org/pdf/2211.13375): the authors translate theoretical results and methods from weak supervision in binary classification to structured prediction (i.e. predicting structured objects, e.g. graphs instead of classes). Our setting involves prediction of classes, not structured objects, so we are unable to compare to this method. However, it is a promising starting point in order to generalize SSME to structured prediction in future work, and we have added a citation to it at Line 579.
> >
> > Suggested baseline 5 (Mandoline, https://arxiv.org/abs/2107.00643): Mandoline is designed for settings which differ from ours in two key respects: 1) Mandoline is used to adjust for distribution shifts between labeled and unlabeled data, but there are no distribution shifts in our setting, and 2) Mandoline assumes access to user-written “slicing functions”, which rely on domain knowledge and are unavailable in many of the settings we study. We are hence unable to compare to Mandoline, but we do agree it is a relevant reference and have added a citation to it at Line 90.
> >
> > **Q1: Evaluating a new, previously unavailable classifier**
> >
> > Given a new classifier, one can refit SSME. Refitting SSME is computationally cheap: for each configuration of (n_labeled, dataset, data_split) we report on in the paper, SSME can be fit in under 5 minutes, using only CPUs. Costs scale as the number of classifiers and examples grows, but are within reason for practical constraints. Alternatively, if one’s estimate of p(y|s) is good given the existing set of classifiers, the additional classifier can be evaluated against labels inferred using the original classifier sets’ predictions. We have clarified this point at Line 248 in the revised draft.
> >
> > **Q2: Role of classifier accuracy and correlation**
> >
> > The more accurate the classifiers are, the more separable the components of the mixture model are. When the classifier set is more accurate, fewer labeled examples are required to yield an accurate estimate of p(y|s). Figure 4 illustrates this phenomenon on synthetic data, and we have clarified this point at Line 525.
> >
> > We make no assumption about the correlation between classifiers – in contrast to prior work – because any assumption is unlikely to hold in practice (predicted probabilities are likely to be correlated). The average correlation between classifiers in our sets for each binary task is 0.53, 0.85, 0.93, 0.81, 0.77 (for ED revisit, critical outcome, hospitalization, toxicity, and SARS-COV inhibition prediction, respectively). This range of values reflects natural correlation between classifiers in practice, since each of our models is either an off-the-shelf classifier or trained using publicly available code. We’ve updated the manuscript with more detail about SSME’s robustness to a range of classifier correlations in Section D.5.

---

> > > ### Author Response · Authors · 2024-11-20
> > > **(continuation of response)**
> > >
> > > **Q3: Sequential availability of classifiers**
> > >
> > > One way to apply SSME is to re-estimate p(y|s) as new classifiers arrive; as mentioned, SSME trains in under five minutes in each individual experiment, making it a feasible procedure to perform multiple times. The possibility of creating a classifier set from classifiers produced during classifier development is an interesting one; SSME could work well for such a setting, and we consider it a valuable avenue for future work.
> > >
> > > Please let us know if our response addresses your primary concerns, or if there is further clarification that would be useful!

---

> > > > ### Author Response · Authors · 2024-11-25
> > > > **Follow-up on response**
> > > >
> > > > Thank you again for your review. Are there any additional concerns you have which our responses above didn’t clarify? If we’ve addressed your concerns, would you consider raising your score?
> > > >
> > > > To summarize our changes in response to your review, we a) have deepened the theoretical grounding of the work by reviewing and summarizing relevant results in the semi-supervised learning literature; b) implemented and compared to the additional relevant baselines you suggested, which our method outperforms; and c) discussed how one might use SSME in settings where classifiers may be sequentially, instead of concurrently, available.

---

### Official Review · Reviewer_hobm · 2024-11-03

**Soundness:** 2
**Presentation:** 3
**Contribution:** 2
**Rating:** 5
**Confidence:** 4

**Summary:**

The paper provides a semi supervised method for evaluation of a set of classifiers. Given a small amount of labeled data, a large amount of unlabeled data on which a set of classifiers can provide scores on any input, estimate accuracies of these classifiers. In particular for unlabeled data $x^1, \dots x^n$ and classifiers $1, \dots m$ , we have scores $s^1, \dots s^n$ where $s^i$ is the set of scores for $x^i$, $s^i = \{s_1^i, \dots s_m^i\}$. For classifier $j$,  $s_j^i$ is the score vector in $\Delta^K$ where $\Delta^K$ is the $K$ dimensional simplex, and $K$ is the number of classes.

Their method comprises on fitting a mixture model that estimates true class probabilities given classifier scores, i.e. learn $P(y | s_1, \dots s_m)$. The mixture model is fit by minimizing a joint distribution over scores and true class probabilities parameterized as
$$P(y, s) = \lambda_L \sum_{i\in D_L} \log P(s^i|y^i) + \sum_{i\in D_U} \log \left(\sum_{k=1}^K P(s^i|y^i = k)P(y^i = k)\right)$$
where $P(s^i|y^i = k)$ is parameterized as
$$P(s | y = k) = \frac{1}{\sum_i P(y^i=k|s^i)} \sum_i \mathcal{K}_h(s - s^i) P(y^i = k |s^i)$$

An EM algorithm is used to estimate $P(y^i = k |s^i)$. In the E step given, the mixture component each data point belongs to, i.e. $P(y^i = k |s^i)$ is estimated by maximizing the objective keeping $\mathcal{K}_h$ fixed, and in the M step, $\mathcal{K}_h$ is optimized keeping $P(y^i = k |s^i)$.

Finally given the fitted $P(y^i | s^i)$, any metric $m$ such as classification accuracy is estimated as  $1/n \sum_{i=1}^n E_{y^i \sim P(y^i | s^i)}[m(s_j^i , y_i)]$.

To evaluate their method, the authors use a 6 publicly available datasets. Their evaluation procedure involves first getting a set of classifiers by training on a split of the data, then estimating $P(y^i | s^i)$ on another estimation split and finally comparing the estimated metric against the true metric from a larger evaluation split of the data.

The authors show that their method performs superior to a number of baselines including labeled, Pseudo labeled, David-Skene, and Bayesian Calibration. Further they provide some ablations on their method 1) when only one instead of many classifier is available, and 2) using a normalization flow based model instead of Kernel Density estimation for the mixture model.

**Strengths:**

1. The presentation of the estimation method given a set of classifiers was clear and easy to understand.

2. The method of fitting a mixture model is sound however not novel.

3. The results are superior compared to baselines.

4. Adequate ablations were performed such as using just one classifier instead of many, providing insights on estimation error with classifier accuracy, and studying the estimation error of method by partitioning into different subgroups.

**Weaknesses:**

1. Under what assumptions does the particular joint model over classifier scores and true latent distribution makes sense is not described.

2. The details regarding how are different classifiers distinct from each other is not given. If I understand correctly, each classifier is trained on the same training data, so I assume the only difference arises from different random initialization ? In that case, it appears that different classifier scores should be very similar to each other. Then the point of using multiple classifiers is not clear.

3. A simple baseline such as majority vote weighted by some function of classifier accuracy could be provided.

3. The exact implementation details of baselines such as David Skene is not provided.

4. Assumption of availability of multiple classifiers appears like a strong assumption.

5. The experiment section could be structured better. Instead of figure 2 and figure 3, tables could have provided more information. Some tables in appendix should be used in the main paper.

**Questions:**

Describe more details on how the baselines were used and how are the set of trained classifiers different from each other.

---

> ### Author Response · Authors · 2024-11-20
> **Thanks for your feedback!**
>
> **Q1: Under what assumptions does the proposed latent variable model make sense?**
>
> Good question. First, SSME assumes that the practitioner has access to continuous scores from multiple classifiers: i.e. the probability outputs from each classifier, as opposed to discretized predictions. SSME’s latent variable model also assumes that the labeled and unlabeled data are drawn from the same distribution. SSME also performs particularly well when certain data and modeling assumptions are met, as noted in our response to Reviewer SyCL. In terms of the available data, if the classifier scores tend to cluster well according to their classes, SSME is likely to outperform using labeled data alone. In terms of modeling, SSME is likely to perform well if the mixture model is well-specified for the true underlying density. Since we use a KDE, our model is flexible to many possible true densities.
>
> In addition to these assumptions, we empirically observe that SSME performs better under certain circumstances. We observe empirically that SSME performs better with a limited number (<=50) of classes, which aligns with the challenges of density estimation in high-dimensional settings (Dinh, Sohl-Dickstein, and Bengio, 2017, Wang and Scott, 2019). Our latent variable model can support higher-dimensional evaluation problems but would require robust semi-supervised density estimation procedures. Furthermore, SSME performs better when the classifier scores are well-separated (according to their classes), i.e. when at least one model has reasonably strong accuracy. While SSME is applicable even when most models are random, evaluation performance may decrease.
>
> We added a paragraph in Section 4 (Line 171) describing these properties.
>
> [Dinh, Sohl-Dickstein, and Bengio, 2017] https://arxiv.org/pdf/1605.08803
> [Wang and Scott, 2019] https://arxiv.org/pdf/1904.00176
>
> **Q2: How are different classifiers distinct from one another?**
>
> Good question –  classifiers within our sets (a majority of which are publicly available pre-trained classifiers) differ in terms of training data, architecture, initialization, and training loss, making our setting faithful to classifier sets a practitioner would be able to acquire. For instance, the seven pre-trained classifiers on the CivilComments dataset differ in their training seeds (0,1, etc.) and training losses (ERM, IRM, GroupDRO, etc.). For the news classification tasks, the pre-trained sentence encoders we use differ by their training data. We include a description of each classifier set in Appendix B.1 and have added a main text reference to this section in Line 265.
>
> **Q3: Comparison to majority vote weighted by classifier accuracy**
>
> Thanks for the suggestion! We added the majority vote baseline, weighted by classifier accuracy, to all  experiments (Figure 2, Tables 1, S3, S4, S5). SSME outperforms the baseline across all experiments, likely because (1) estimates of classifier accuracy can be noisy (since they necessarily rely only on the labeled data sample) and (2) discretizing classifier predictions through a vote removes information that is valuable for inferring the true label for an unlabeled example.
>
> **Q4: Exact implementation details for baselines, including Dawid-Skene.**
>
> Thank you for the suggestion! We include exact implementation details for each baseline in Appendix B.2; we have added a main text reference to this section in line 310. In short, we use a public implementation of Dawid-Skene with the default parameters (tolerance of 1e-5 and 100 iterations of EM) and apply the method to a matrix of discretized classifier predictions. We also use public implementations of AutoEval & Bayesian-Calibration, and implement Pseudo-Labeling using a logistic regression with default parameters.
>
> **Q5: How realistic is the assumption of multiple classifiers?**
>
> Many real-world settings have multiple available classifiers, including text classification, medical imaging and diagnostics, and protein property prediction. For instance, the OpenLLM leaderboard on HuggingFace lists four language models with >50% accuracy on MMLU-Pro (a standard multiple choice dataset), and may be difficult to know a priori which one is best for a particular downstream task absent labeled data. Of note, 25 of our 42 models are publicly available pre-trained models, while the remaining models were all produced using publicly available model training code, testifying to the realism of our settings. Finally, although SSME has particular advantages when using multiple classifiers, it can also work well with a single classifier (SSME-KDE-M in Tables S3, S4), often outperforming the baselines.
>
> **Q6: Including tables in main text**
>
> Thanks for the suggestion. We’ve moved the mean absolute error results (with respect to accuracy) into the main text as Table 1, in line with your suggestion.
>
> Please let us know if our response addresses your primary concerns, or if further clarification would be useful!

---

> > ### Comment · Reviewer_hobm · 2024-11-20
> > **Answers not clear**
> >
> > 1. The assumptions under which latent variable model makes sense are still not clear. "we assume that the unlabeled samples s(i) are drawn from the same distribution as the labeled samples (y(i), s(i))." If I understand correctly, s(i) are classifier scores and not the unlabeled samples.
> >
> > 2. (Q2: How are different classifiers distinct from one another?) How are they different in terms of their outputs, i.e. do they make similar or distinct predictions in different subsets of the input space ?
> >
> > 3. (Comparison to majority vote weighted by classifier accuracy) : The baseline is weak, one should consider the weights as some hyperparameters and tune them, for e.g. take weights to be some parameterized function of accuracy and then tune those parameters.
> >
> > 4. The assumption on access to multiple classifiers still look artificial to me specially in the setting where one is concerned with evaluation of a classifier.

---

> > > ### Author Response · Authors · 2024-11-21
> > > **Clarifications and additional results**
> > >
> > > Thank you for the swift response and for continuing to engage with our work! We respond to your additional questions below:
> > >
> > > **Q1: The assumptions under which latent variable model makes sense are still not clear. "we assume that the unlabeled samples s(i) are drawn from the same distribution as the labeled samples (y(i), s(i))." If I understand correctly, s(i) are classifier scores and not the unlabeled samples.**
> > >
> > > Thank you for this question. You are correct in that $\mathbf{s}^{(i)}$ refers to the classifier scores. We assume the joint distribution p(y, s) remains constant across both the labeled and unlabeled data. For the labeled data we observe both y and s; for the unlabeled data we observe only s.
> > >
> > > **Q2: (Q2: How are different classifiers distinct from one another?) How are they different in terms of their outputs, i.e. do they make similar or distinct predictions in different subsets of the input space ?**
> > >
> > > Thank you for clarifying!
> > >
> > > Our classifier sets vary in the similarity of classifier predictions to one another. Across the five binary datasets we examine, the average correlation between classifier predicted probabilities varies significantly: 0.93, 0.85, 0.53, 0.93, 0.77 for hospitalization, critical outcome, ED revisit, toxicity, and SARS-COV inhibition prediction, respectively (detailed in Section D.5).  To respond to your question, we additionally conducted an analysis of how often the classifiers disagree in terms of their thresholded predictions. The percentage of examples which receive different predictions (thresholded at 0.5) is 23%, 4%, 0%, 22%, 9% for hospitalization, critical outcome, ED revisit, toxicity, and SARS-COV inhibition prediction, respectively. (The reason the classifiers never disagree on the ED revisit dataset is that the dataset has a very low base rate, so each classifier predicts each example to be negative when we use a threshold of 0.5).  Please let us know if there are additional details that would be helpful!
> > >
> > >
> > > **Q3: (Comparison to majority vote weighted by classifier accuracy) : The baseline is weak, one should consider the weights as some hyperparameters and tune them, for e.g. take weights to be some parameterized function of accuracy and then tune those parameters.**
> > >
> > > Thank you! In response to your follow-up comment, we added this additional baseline. Specifically, we weighted each classifier by a quadratic function of classifier accuracy, where we learn the parameters using the available labeled data. We find that SSME outperforms this baseline as well.
> > >
> > > We have implemented both variants of the requested baseline – where each classifier is weighted by a classifier accuracy, and where each classifier is weighted by a learned function of classifier accuracy – and find that SSME outperforms both. We are happy to add these results to the draft as well if you think it would be helpful!
> > >
> > > **Q4: The assumption on access to multiple classifiers still look artificial to me specially in the setting where one is concerned with evaluation of a classifier.**
> > >
> > > Thank you for clarifying. Note that SSME is not restricted to settings with multiple available classifiers. SSME can be used to evaluate single classifiers at a time (where $\mathbf{s}^{(i)}$ is the output of a single classifier, instead of multiple). Our experiments show it can also work well with a single classifier (SSME-KDE-M in Tables S3, S4).
> > > Instead, we show that there are many settings where multiple classifiers may be available and it is unclear how to choose the best one a priori, especially without access to significant labeled data.  Such settings are common due to the rising popularity of model hubs (Pal et al., 2024, Taraghi et al., 2024, You et al., 2022) and include image classification, content moderation, and healthcare, rendering SSME widely applicable. In such settings, SSME simultaneously evaluates all available classifiers, unlike other baselines which can only evaluate single classifiers at a time. This is a comparative advantage of SSME: by aggregating  predictions across multiple classifiers, we can get better estimates of the true label than from single classifiers alone.
> > >
> > > Pal et al., 2024 - https://arxiv.org/abs/2403.02327
> > >
> > > Taraghi et al., 2024 - https://arxiv.org/pdf/2401.13177
> > >
> > > You et al., 2022 -https://www.jmlr.org/papers/v23/21-1251.html

---

> > > > ### Author Response · Authors · 2024-11-25
> > > > **Follow-up on response**
> > > >
> > > > Thank you again for your review and for following up with additional questions regarding our submission. Are there any additional concerns you have which our responses above didn’t clarify? If we’ve addressed your concerns, would you consider raising your score?
> > > >
> > > > To summarize our changes in response to your comments, we have a) added two majority vote baselines as requested in your review and follow-up question, which our method outperforms; b) clarified how the classifiers are distinct from each other (including how often they disagree); and c) clarified under what assumptions SSME works well.

---

> > > > > ### Comment · Reviewer_hobm · 2024-12-02
> > > > >
> > > > > Thank you for making an attempt to address the concerns. However I still think the proposed method is not novel enough to recommend acceptance, and I am not convinced about the practical utility of the setting. Therefore I would like to keep my current score.

---

> > > > > > ### Author Response · Authors · 2024-12-02
> > > > > > **Response on novelty & practical utility**
> > > > > >
> > > > > > Thanks for your response, and for your detailed engagement with our work! We would like to clarify that the novel contribution in our work is a new method, SSME, for evaluating multiple classifiers using labeled and unlabeled data (as opposed to the approach to fitting the mixture model, expectation-maximization, which as you note is of course not novel). SSME is the **first** evaluation method to simultaneously learn from: (i) multiple machine learning classifiers, (ii) probabilistic predictions over all classes, and (iii) unlabeled data. No reviewer has identified a baseline equivalent to SSME, which outperforms known baselines.
> > > > > >
> > > > > > We respectfully disagree about the practical utility of the setting SSME addresses: i.e., evaluating multiple classifiers with limited labeled data. Many works focus on this setting [1, 2, 3, 4], and the popularity of model hubs like HuggingFace have made problems of classifier evaluation – particularly in the presence of multiple classifiers – a common challenge. Moreover, we have clarified that SSME is not limited to settings with multiple classifiers, and can perform quite well with one.
> > > > > >
> > > > > > Thank you again for your detailed engagement with our paper!
> > > > > >
> > > > > > [1] https://arxiv.org/abs/2401.13177
> > > > > > [2] https://arxiv.org/abs/2110.10545
> > > > > > [3] https://arxiv.org/abs/2306.03900
> > > > > > [4] https://arxiv.org/abs/2210.09236

---

### Official Review · Reviewer_aeyD · 2024-11-04

**Soundness:** 3
**Presentation:** 3
**Contribution:** 3
**Rating:** 8
**Confidence:** 5

**Summary:**

The paper introduces Semi-Supervised Model Evaluation (SSME), a novel method designed to evaluate machine learning classifiers using both labeled and unlabeled data, addressing the challenge of obtaining large labeled datasets. SSME leverages the availability of multiple classifiers, continuous classifier scores, and abundant unlabeled data to estimate the joint distribution of true labels and classifier predictions through a semi-supervised mixture model.  The authors validate SSME across four challenging domains—healthcare, content moderation, molecular property prediction, and image annotation—showing that it significantly reduces error rates in performance estimation compared to traditional methods.

**Strengths:**

Originality: SSME is a pioneering approach that creatively combines labeled and unlabeled data for model evaluation, filling a critical gap in existing methodologies.
Clarity: The abstract effectively communicates the core concepts and findings, making it accessible to a broad audience while maintaining technical depth.
Significance: This work addresses a significant challenge in machine learning evaluation, potentially benefiting many fields where labeled data is scarce. Its implications for improving model assessment and performance understanding are substantial.

**Weaknesses:**

In fact, there has been a lot of work in recent years on evaluating language models, but none of these are mentioned in this paper. Moreover, since the datasets and classifiers selected for the experiment also involve the architecture of large language models, these methods of evaluating language models are not mentioned in the comparison method.

**Questions:**

What is the complexity of the method in this paper? In other words, is this approach worth the computational resources we consume compared to tolerating the errors that come with traditional evaluation methods?

---

> ### Author Response · Authors · 2024-11-20
> **Thanks for your feedback!**
>
> Thanks for your positive review! We are glad to hear you found the paper clear and creative, and the contribution significant. We have improved the draft in response to your suggestions (detailed below), and welcome any further feedback.
>
> **Q1: Discussion of recent efforts to evaluate large language models.**
>
> A1 Thanks for the useful suggestion – we’ve expanded our discussion of model evaluation to include recent approaches to evaluating LLMs (Lines 889 - 893). These approaches differ from our own because they are LLM-specific while our method is not; we have included citations to this other work to contextualize our contribution.
>
> **Q2: Computational complexity of the proposed method**
>
> Good question. SSME’s computational complexity depends on the model class used to parameterize the mixture model; for example, when parameterized by a KDE, computational complexity grows quadratically in the number of examples (although there are ways to improve efficiency through, for example, binning [1]). In practice, using SSME to evaluate classifiers in our setting is not computationally costly: we fit SSME to each configuration of (n_labeled, dataset, data_split) reported in the paper in under five minutes.
>
> [1] http://www-stat.wharton.upenn.edu/~lzhao/papers/MyPublication/Fast_jcgs.2010.pdf

---

> > ### Comment · Reviewer_aeyD · 2024-11-29
> >
> > Thanks for the author's clarification. I'm going to keep my positive rating on this paper.

---

### Author Response · Authors · 2024-11-20
**Thanks for your feedback!**

We thank all reviewers for their constructive feedback, and for their recognition of the importance of the problem we address – measuring model performance with little labeled data. We appreciate their saying that we “fill a critical gap,” “address a common bottleneck,” and “introduce a setting often encountered in practice.” The reviewers also seem convinced that we improve over the 5 baselines we compare to: one reviewer states that the results are "superior compared to baselines," another notes that our method "significantly reduces error rates... compared to traditional methods" and a third notes that the "results indicate improvements over baselines."

The reviewers also provided a number of useful suggestions for improving the manuscript, all of which we have addressed by performing additional experiments and submitting a revised draft. (Note that the updated draft appears to be over the page count because we used track changes to make clear what the edits are, but it is within the page count.) We summarize our major changes below and provide additional details in the point-by-point response to each reviewer.

* Reviewers suggested comparisons to additional baselines. We have reviewed all suggested baselines, and added comparisons to the 2 additional baselines which are comparable to our method and not previously included in our evaluation. Our method outperforms these baselines. We have also clarified that several suggested baselines are already included in our evaluation.
* Reviewer SyCL requested discussion of the theoretical properties of our method; we have added this in Section E.3 by drawing from the semi-supervised mixture model estimation literature.
* While several reviewers found the work clear and easy to understand, others suggested details to clarify which we have now addressed, including how classifiers within a set differ from one another (Section B.1), connections to LLM evaluation (Lines 889 - 893), why we model the distribution of classifier scores *s* instead of inputs *x* (Lines 164-170), and our procedure for inferring ground truth labels (Lines 240-246). In particular, this addresses the main concern of reviewer w43E, which was lack of methodological clarity.
* Reviewers asked about the scalability of our method. We have provided additional details on the method's computation time: SSME fits all configurations of (n_labeled, dataset, data split) reported in the paper in under 5 minutes, creating minimal computational overhead. We have also clarified that the scalability of the method depends on the model class used to parameterize the mixture model; hence, for extremely large datasets, efficient model classes (e.g., fast implementations of KDEs) could be used.

We believe the updated manuscript to be much stronger because of the reviewers’ feedback and look forward to continued discussion.

---

### Meta-Review · Area_Chair_Eeby · 2024-12-24

**Metareview:**

Summary

The paper proposes a semi-supervised evaluation method for classifier performance using unlabelled data, addressing the lack of annotated test datasets. It introduces a semi-supervised mixture model to estimate the joint distribution of ground truth labels and classifier predictions. The method uses continuous classifier scores, multiple classifiers, and abundant unlabeled data to infer both true labels and classifier performance. Empirical studies across four domains show that this approach improves evaluation accuracy.

Strengths


The paper addresses a practical and underexplored problem, making it highly relevant for real-world scenarios where labelled test data is unavailable. Its strengths include strong experimental results that validate the proposed method and its application across multiple domains, demonstrating robustness and generalizability. The paper also offers a scalable solution.

Weaknesses

The main concern is that the method may lack novelty, as it appears to be generative modelling through a mixture model to estimate the joint probability of the ground truth labels of unlabelled data and the predictions made by classifiers. The assumption underlying the joint modelling of classifier scores and ground truth labels is not guaranteed to hold universally, potentially limiting its applicability.

Reasons for decision

Overall, the primary concern with the paper lies in its lack of novelty. Neither the paper itself nor the rebuttal successfully convinced reviewers of its methodological originality, as the techniques employed are perceived as a combination of existing methods, albeit with different objectives. While the paper explores an interesting and potentially impactful direction, addressing the challenge of insufficient labelled test data, it currently falls short in practicality due to concerns about its assumptions and technical implementation. These issues, highlighted by reviewers, need to be thoroughly addressed in future versions of the work.

**Additional Comments On Reviewer Discussion:**

During the discussion phase, the paper faced mixed feedback. Reviewer hobm consistently argued for rejection, citing concerns about the paper's lack of novelty and insufficient theoretical motivation. This opinion was supported by Reviewer w43e, who agreed that the method does not demonstrate significant.

Reviewer aeyD maintained a high score of 8, providing a brief review without raising significant concerns and demonstrating high confidence in their evaluation. In contrast, Reviewer hobm detailed the paper’s technical aspects, focusing on assumptions, technical details, and empirical studies. While some of their concerns were partially addressed with revisions, questions about assumptions and practical applicability remained unresolved, leading them to hold their original score. Reviewer SyCL rated the paper a 5, acknowledging its simplicity, novelty, and good empirical performance but raising concerns about the lack of robust theoretical analysis and certain assumptions. The authors' rebuttal on theoretical aspects was directional rather than concrete, which was understandable given the limited time, but this did not lead to further engagement from this reviewer. Reviewer w43e provided an in-depth review, summarizing technical details and expressing concerns about novelty. After revisions, they increased their score to 6 but reiterated during discussions that the method lacks sufficient novelty.

In response to these concerns, the authors revised the paper to include additional empirical studies with new baselines, discussions on scalability, and improved clarity regarding theoretical properties. However, these revisions did not fully address the concerns about assumptions or theoretical soundness.

In summary, the paper received a mixed evaluation with one score of 8 (brief, positive review), two scores of 5, and one score of 6. The longer, more detailed reviews highlighted concerns about assumptions, novelty, and theoretical motivation. Despite the interesting direction and potential practical value, the lack of originality and reliance on assumptions were recurring themes, leaving three reviewers unconvinced of the paper’s merit for acceptance.

---

### Decision · Program_Chairs · 2025-01-22

Reject